# Epigenetic regulation of spurious transcription initiation in *Arabidopsis*

Ngoc Tu Le [1], Yoshiko Harukawa[1], Saori Miura[1], Damian Boer [2], Akira Kawabe [3] & Hidetoshi Saze [1✉]

In plants, epigenetic regulation is critical for silencing transposons and maintaining proper gene expression. However, its impact on the genome-wide transcription initiation landscape remains elusive. By conducting a genome-wide analysis of transcription start sites (TSSs) using cap analysis of gene expression (CAGE) sequencing, we show that thousands of TSSs are exclusively activated in various epigenetic mutants of *Arabidopsis thaliana* and referred to as cryptic TSSs. Many have not been identified in previous studies, of which up to 65% are contributed by transposons. They possess similar genetic features to regular TSSs and their activation is strongly associated with the ectopic recruitment of RNAPII machinery. The activation of cryptic TSSs significantly alters transcription of nearby TSSs, including those of genes important for development and stress responses. Our study, therefore, sheds light on the role of epigenetic regulation in maintaining proper gene functions in plants by suppressing transcription from cryptic TSSs.

---

[1] Plant Epigenetics Unit, Okinawa Institute of Science and Technology (OIST), 1919-1 Tancha, Onna-son, Kunigami-gun, Okinawa 904-0495, Japan. [2] Wageningen University & Research, Droevendaalsesteeg 4, 6708 PB Wageningen, Netherlands. [3] Faculty of Life Sciences, Kyoto Sangyo University, Kyoto 603-8555, Japan. ✉email: hidetoshi.saze@oist.jp

Eukaryotic genomes are comprised a large part of mobile genetic sequences, so-called transposable elements (TEs)[1]. Due to their mobility, TEs induce various alterations to the host genome, ranging from genetic mutations to large-scale genomic rearrangements, such as inversions and translocations[2,3]. Genetic variations caused by TEs can introduce novel regulatory elements and therefore be a major driving force underlying genome evolution[1,2]. On the other hand, uncontrolled activities of TEs can severely damage gene expression and the integrity of the host genomes[3].

To suppress negative impacts without losing potential benefits brought in by TEs, both plants and animals have evolved numerous epigenetic mechanisms involving DNA methylation, histone modifications, and small non-coding RNAs, allowing TEs remain silenced in their genomes[4,5]. Compared to mammals, plants are equipped with a different set of epigenetic mechanisms for greater adaptability to dynamic environmental changes, partly due to their sessile nature. For example, in mammalian genomes, DNA sequences are mainly methylated at the cytosine in the CG dinucleotides, while in plants cytosine methylation exists in both CG and non-CG contexts, which has different functional impacts on gene and TE regulation[6].

In the plant model *Arabidopsis thaliana* (*A. thaliana*), DNA methylation is established de novo by the RNA-directed DNA methylation (RdDM) pathway, which requires the functional activity of PolIV and PolV, two plant-specific RNA polymerases[5]. After establishment, methylation patterns can be maintained by different factors depending on cytosine contexts. CG methylation is maintained by METHYLTRANSFERASE 1 (MET1), a plant homologue of the mammalian DNA (cytosine-5)-methyl-transferase 1 (DNMT1). Maintenance of DNA methylation in CHG context, on the other hand, is facilitated by CHROMO-METHYLASE 3 (CMT3) in a positive feedback loop with the histone H3K9 methylase KRYPTONITE (KYP) (or SUPPRESSOR OF VARIEGATION 3-9 HOMOLOGUE 4 (SUVH4))[7]. Together with two of its paralogues, SUVH5 and SUVH6, KYP regulates the genome-wide accumulation of H3K9me2 and consequently, CHG methylation[6]. CHH methylation can be maintained by either CMT2 or DOMAINS REARRANGED METHYLASE 2 (DRM2) depending on the features of their targets, in which DRM2 often methylates short, euchromatic TEs, while CMT2 targets long TEs located in histone H1-containing heterochromatic regions with the help of chromatin remodeler DECREASED DNA METHY-LATION 1 (DDM1)[8]. These epigenetic pathways in plants, however, are highly interwoven. For example, MET1 and CMT3 are involved in maintaining asymmetric methylation, while DMR2 and CMT2 may also affect DNA methylation in other contexts[9].

Epigenetic silencing of TEs inevitably confers regulatory impacts on gene expression, especially when TEs are located close to transcription units[2,4]. In plants, repressive modifications triggered by TE insertions within introns or promoter regions can attenuate or even turn off the expression of the associated genes[10–12]. At a global scale, genes harboring, or located close to, silenced TEs exhibit lower expression than their counterparts[13,14]. Due to such unfavorable impacts, plants have evolved specific pathways to keep transcription units clear of repressive modifications, or to tolerate the presence of such modifications when necessary. For example, in *A. thaliana* the Jumonji C (jmjC) domain-containing histone demethylase INCREASE IN BONSAI METHYLATION 1 (IBM1) prevents repressive H3K9 methylation and consequently, CHG methylation, from accumulating at actively transcribed genes[15]. On the other hand, host factors, such as INCREASE IN BONSAI METHYLATION 2 (IBM2) and Enhanced Downy Mildew 2 (EDM2) are required for proper transcription of genes containing heterochromatic domains[16,17], likely due to the functional importance of these domains[14].

The development of high resolution 5′ end-centered expression profiling techniques, such as oligo-capping methods[18] or cap analysis of gene expression (CAGE)[19], has greatly advanced our understanding of gene regulation at a transcription initiation level. Studies employing these techniques have revealed both common and distinct features of the core promoters and their origin and regulation, in many organisms[20–22]. In mammals, for example, CAGE sequencing (CAGE-seq) analyses revealed that a large fraction of cell-type specific transcripts in stem and cancer cells originate from long terminal repeats (LTRs) of retroelements[23,24]. The loss of DNA methylation also causes spurious transcription within thousands of genes in mouse embryonic stem cells[25]. In addition, modulating DNA methylation and histone deacetylation pathways pervasively activates cryptic transcription start sites (TSSs) normally silenced in human cells[26]. These examples demonstrate the importance of epigenetic mechanisms in regulating transcription initiation in mammalian genomes.

In plants, large-scale analyses have determined thousands of TSSs, providing fundamental information about genetic structure and regulatory elements important for transcription in plant genomes[27,28]. Previous studies have also revealed core promoter structures and sequence elements associated with plant TSSs[29–31]. However, these studies mainly focus on active TSSs in the wild-type background. The contribution of epigenetic regulation to shaping the genome-wide transcription initiation landscape and its functional significance in plants, therefore, remains largely unexplored.

To dissect the functional impacts of epigenetic regulation in shaping the plant transcription initiation landscape, we employ CAGE-seq to generate the genome-wide profiles of TSSs at a high resolution for various mutants of *A. thaliana* that compromise epigenetic control. Our analysis identifies thousands of TSSs exclusively activated in the mutant backgrounds, demonstrating that epigenetic regulation profoundly affects transcription initiation in *Arabidopsis*. These so-called cryptic TSSs are mainly located at heterochromatic regions, which hinder their accessibility to RNA Polymerase II (RNAPII) transcription machinery. The alteration of DNA methylation maintenance in *met1* activates the largest number of cryptic TSSs, which significantly overlap with the targets regulated by other epigenetic pathways. A large fraction of cryptic TSSs originate from TEs of both retro and DNA-transposon families, suggesting that TEs are reservoirs of putative TSSs in the *A. thaliana* genome. Strikingly, the activation of cryptic TSSs significantly alters the regular transcription of nearby TSSs, which includes those of genes important for development and stress responses in *Arabidopsis*. This study, therefore, sheds light on the role of epigenetic regulation in maintaining proper gene functions in plants by suppressing transcription initiated from cryptic TSSs. In addition, the accompanying data are a valuable resource for studying the epigenetic control of the transcription of genes and TEs in plants.

## Results

**Mapping TSSs in epigenetic mutants of *A. thaliana* by CAGE-seq.** To gain a comprehensive view regarding the epigenetic regulation of transcription initiation in plants, we performed CAGE-seq analyses of various *A. thaliana* mutants, where epigenetic control is compromised, including mutants of maintenance DNA methyltransferase *met1*, the chromatin remodeler *ddm1*, RdDM pathway components *nrpd1* and *nrpe1*, histone H3K9 methyltransferases *suvh456*, histone H3K9 demethylase *ibm1*, and intragenic heterochromatin regulatory factors, *ibm2* and *edm2*. A total of 1,250,203,294 CAGE-seq reads were mapped to the *A. thaliana* Col reference genome, achieving an average

mapping efficiency of 97.53%. Of which, 402,814,394 reads were mapped uniquely, compiling a large collection of CAGE-seq data for this model plant (Supplementary Data 1).

The expression of individual CAGE-based TSSs (CTSSs) was highly correlated between replicates (the median of Pearson correlation coefficients was 0.95) (Supplementary Fig. 1a, b), confirming the reproducibility of our data. In total, 37,726 consensus tag clusters representing single TSSs were identified across all samples (hereafter TSSs is used to refer to consensus tag clusters identified in this study, to distinguish from the TAIR-annotated TSSs), of which about 30% were exclusively expressed in the mutant backgrounds (Supplementary Data 2).

To confirm the relevance of our data, we analyzed the genome distribution of 26,561 TSSs identified in wild-type sample. A majority of them (18,634 or ~70%) were located in promoters and 5′ UTRs of 17,722 (~64%) annotated genes (Fig. 1a), and about one-fourth (~24%) were located in intragenic regions, of which exonic TSSs were more prevalent than the intronic counterparts. Although the mechanisms leading to the prevalence of exonic TSSs in the plant genomes have yet been clear[21], a part of them may represent 5′-end capped products of post-transcriptional processing of mature mRNAs, as described in human and vertebrate genomes[32,33]. Alternatively, some may correspond to cryptic promoters that trigger spurious transcription from gene bodies[25,34], or to mis-annotated TSSs[22]. Nevertheless, consistent with a previous study[21], the expression of intragenic TSSs was significantly lower than that of their counterparts located in promoters and 5′ UTRs (Fig. 1b). Moreover, the TSSs in promoters and 5′ UTRs were found in close proximity to the TAIR10-annotated TSSs (Supplementary Fig. 2a, b). A similar result was obtained using the Araport11 genome annotations, (Supplementary Fig. 3a–c), with a shift in the numbers of TSSs assigned to each genome feature (Fig. 1a, Supplementary Fig. 3a). Because of the higher consistency with the TSSs identified by our CAGE-seq (Supplementary Figs. 2a, b, 3b, c), TAIR10 annotations were used in further downstream analysis. On the other hand, active genes supported by CAGE and mRNA-seq were largely overlapped (Supplementary Fig. 2c), suggesting that active transcription events in *A. thaliana* can be efficiently captured by our CAGE-seq data.

We then compared wild-type TSSs identified by CAGE-seq with those reported by the paired end analysis of transcription start site (PEAT) method[31]. They were indeed consistent even though the samples were prepared from different tissues (Supplementary Fig. 3d–f). At a local scale, the promoter architecture of two well-studied genes, *ALMT1* (*AT1G08430*) and *sAPX* (*AT4G08390*), was also reexamined. The former has three functional TSSs within its promoter and the latter has one upstream and one intragenic TSS[21]. Our data recapitulated these structures (Supplementary Fig. 4a, b), confirming its consistency with previous studies[21,31].

It has been found that the loss of CG methylation at a SINE-related repeat in the promoter region triggered the ectopic expression of the homeobox gene *FLOWERING WAGENINGEN* (*FWA*), causing a late flowering phenotype of *Arabidopsis*[11,35,36]. CAGE-seq analysis identified a TSS encoded within the SINE repeat, which was highly activated in *met1* and *ddm1* backgrounds (Fig. 1c). In addition, the ectopic activation of the TSS of the F-box gene *SUPPRESSOR OF drm1 drm2 cmt3* (*SDC*), whose promoter contains a tandem repeat co-regulated by H3K9 methylation and the RdDM pathway[8], was also detected by our data (Fig. 1c).

Taken together, these results demonstrate that our CAGE-seq data can be effectively exploited for the detection and analysis of both regular and cryptic TSSs under epigenetic control.

**Modulating epigenetic regulation activates many cryptic TSSs.** Next, we investigated the impact of epigenetic regulation on the transcription initiation landscape in the *A. thaliana* genome in greater details. Compromising epigenetic controls significantly affected the transcription initiated from hundreds to thousands of TSSs, in which the defect of the maintenance DNA methylation pathway in *met1* induced changes at the largest number of targets (Fig. 2a), followed by *ibm1*, *ddm1*, *suvh456*, and *pol4*. To our surprise, *ibm2* and *edm2*, which cause the transcriptional defect of *IBM1*[16,17], had a lower number of affected TSSs than *ibm1*, suggesting that the IBM1 function is partially maintained in these mutants.

Of the altered TSSs, many were activated de novo in the mutant backgrounds and were not associated with any TAIR10-annotated TSSs (Fig. 2a, Supplementary Fig. 2b, Supplementary Data 3). They were also largely distinct from the TSSs reported by PEAT-seq[31] and the TSSs identified in multiple tissues and light stress conditions in *A. thaliana*[21] (Supplementary Fig. 5a, b), suggesting that they are cryptic TSSs suppressed by epigenetic mechanisms (referred herein as EPICATs, for EPigenetically Induced Consensus tAg clusTers). Our data showed that the EPICATs activated in *met1* largely overlapped with the EPICATs regulated by other mutants, confirming the profound regulatory impact of MET1 on the genome-wide transcription initiation in *A. thaliana* (Fig. 2b). On the other hand, *ddm1* and RdDM-associated mutants (*pol4* and *pol5*) induced stronger activation of the EPICATs than *met1* (Fig. 2c). Due to the minor numbers of instances, targets of *ibm2* and *edm2* were excluded from further analysis. Similar results were obtained using the Araport11 annotations (Supplementary Figs. 3c, 5c), confirming the robustness of our analysis.

As the transcription orientation at regulatory regions of eukaryotes can be either unidirectional[20] or bidirectional[37], we examined the directionality of transcription initiated at EPICATs. Our data showed that transcription at the EPICATs in *met1* was mainly uni-directional, similar to that of the TAIR10-annotated TSSs in *A. thaliana* (Supplementary Fig. 6a[20]). Moreover, the expression levels of EPICATs were not significantly different from those of the annotated TSSs activated de novo in epigenetic mutants (Supplementary Fig. 6b). We also found that, tag clusters corresponding to the EPICATs mainly had narrow peaks (NPs), especially those activated in *ddm1*, *met1*, and *pol5* (Supplementary Fig. 6c), suggesting that they may have a well-defined underlying genetic architecture[31,38].

To elucidate putative mechanisms regulating the activity of EPICATs, we first examined the genomic regions where they reside. EPICATs were mainly located at intergenic regions, except the EPICATs in *ibm1*, of which a majority were intragenic (Fig. 3a, Supplementary Fig. 6d). These intragenic EPICATs, however, may not be directly regulated by the activity of IBM1, because they were not associated with increased CHG methylation in the *ibm1* background (Fig. 3b). In contrast, the EPICATs in other mutants were located in genomic regions decorated with repressive chromatin modifications, such as DNA methylation, H3K9me2, and H3K27me1 (Supplementary Fig. 7a, b). Compared to the EPICATs in other mutants, those activated in *pol4* and *pol5* were also associated with a higher level of CHH methylation and 24 nt siRNAs, the hallmarks of the RdDM pathway (Supplementary Fig. 7a, c). Moreover, DNA methylation at the EPICATs in all mutants, except in *ibm1*, was significantly reduced, in concomitant with their activation (Fig. 3b, Supplementary Fig. 7d), suggesting that in wild-type plants transcription initiation at EPICATs is directly suppressed by repressive epigenetic modifications.

Since heterochromatic modifications, such as DNA methylation and H3K9me2, are often associated with closed chromatin in

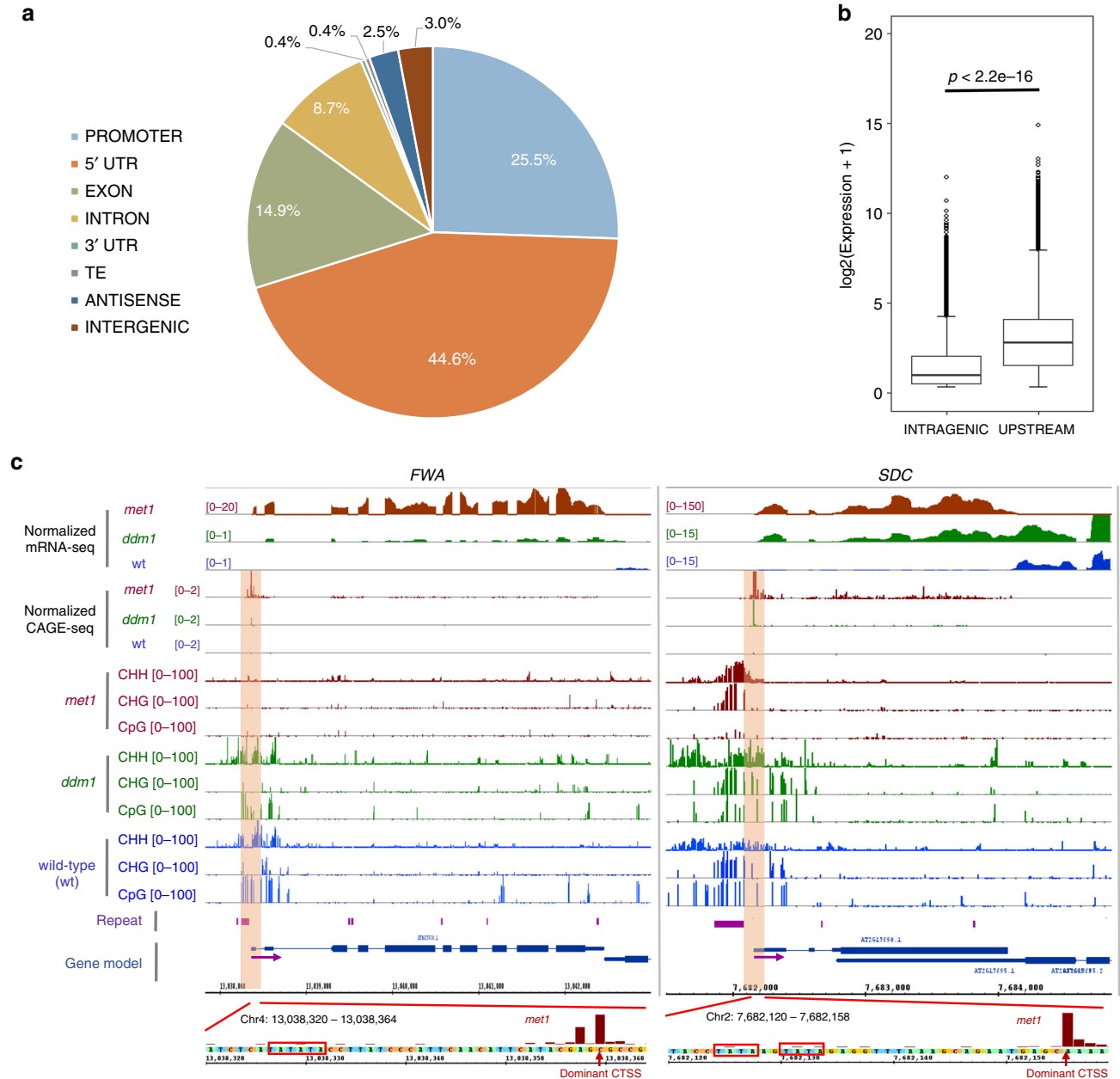

**Fig. 1 Characterizing the TSSs identified in wild-type *A. thaliana* plant by CAGE-seq. a** Genome-wide distribution of TSSs ($n = 26{,}561$) identified in wild-type *A. thaliana*. Genomic features were used for overlapping test in the following order: PROMOTER > 5′ UTR > 3′ UTR > INTRON > EXON > ANTISENSE > TE > INTERGENIC. **b** A boxplot showing that intragenic TSSs are less expressed than their counterparts located in the upstream regions (promoters or 5′ UTRs). *p*-value was calculated by two-sided Mann–Whitney test on the expression of the TSSs. The centerline in the plot represents the median. The bounds of the box are the first and third quartiles (Q1 and Q3). Whiskers represent data range, bounded to 1.5 * (Q3-Q1). Points outside this range are represented individually by hollow circles. **c** Browser tracks showing the ectopic activation of the TSSs at *FWA* (*AT4G25530*) and *SDC* (*AT2G17690*) gene loci in the *met1* and *ddm1* backgrounds (indicated by orange windows). Their dominant CTSSs (red arrows) were aligned to the upstream sequences, indicating the presence of TATA-box motifs (red boxes) encoded within (left panel) or downstream (right panel) of the nearby repeats. Purple arrows indicate the direction of transcription.

plant genomes[39], their loss may alter the access to genomic regions harboring EPICATs. We therefore examined how the accessibility of these loci changes in the mutant backgrounds. For this purpose, the EPICATs activated in *ddm1* were used as a proxy due to the large number of instances and the availability of public data characterizing chromatin openness in *ddm1*[40]. Indeed, chromatin around the EPICATs became highly accessible in *ddm1*, compared to wild-type plants, as measured by the sensitivity to DNaseI (Fig. 3c). Furthermore, ChIP-seq analysis showed that RNAPII phosphorylated at Ser5 (Ser5P) and Ser2

(Ser2P) in the C-terminal domain (CTD), the hallmarks of transcription initiation and elongation[41] respectively, were also highly accumulated at the EPICATs in most mutant backgrounds (Fig. 3d, Supplementary Fig. 7e). These data demonstrate that repressive chromatin suppresses the activity of EPICATs by preventing the access of transcription machinery to genomic regions encompassing potential TSSs.

Ectopic transcription initiation in mutants and the convergence of various epigenetic pathways on a large number of EPICATs (Fig. 2b), together with the narrow shapes of tag

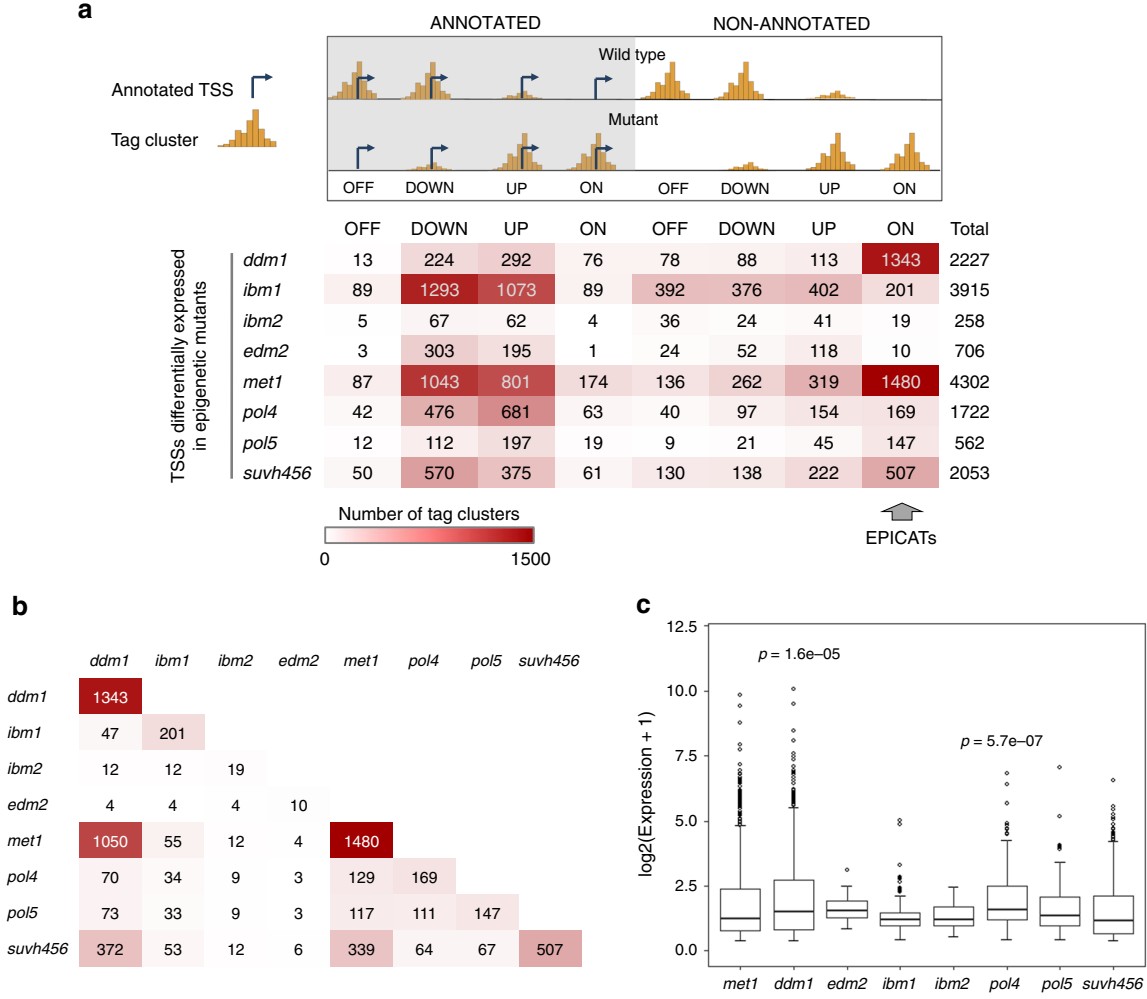

**Fig. 2 Modulation of epigenetic control has profound impacts on transcription initiation in *Arabidopsis*. a** Classification of the TSSs differentially expressed in epigenetic mutants. A TSS was defined as ANNOTATED if the distance from its dominant CTSS to the nearest TAIR10-annotated TSS in the same orientation is less than 180 nt, or NON-ANNOTATED if not. NON-ANNOTATED TSSs activated de novo in the mutant backgrounds are named EPICATs (standing for EPigenetically Induced Consensus tAg clusTers). **b** Convergence of different epigenetic pathways in regulating EPICATs. Each cell presents the number of EPICATs commonly regulated in the mutants indicated in the corresponding row and column. The color code is the same as in **a**. **c** Impacts of different epigenetic pathways on EPICAT expression. Comparisons were conducted against the expressions of the EPICATs in *met1* by two-sided Mann–Whitney test. *p*-values < 0.01 are shown.

clusters corresponding to most of the EPICATs (Supplementary Fig. 6c), suggest that these loci harbor functional genetic features, such as promoter structure and/or regulatory sequences[21], in addition to repressive chromatin modifications. Therefore, genetic sequences surrounding EPICATs were analyzed. Interestingly, DNA elements and motifs enriched around EPICATs exhibited spatial architecture similar to that of regular plant promoters[20,30], with a sharp accumulation of TATA-box at 36 nt upstream and CA-rich/CT-rich (Y-patch) motifs around the TSSs (Fig. 3e, Supplementary Fig. 8). TATA-box, a core promoter motif conserved in both plants and animals[30,38], was especially enriched at the EPICATs in *met1* and *ddm1*. The enrichment of the Telobox motif (AAACCCTA), which is known to recruit development-associated repressive modification H3K27me3 in *A. thaliana*[42], was also found at the EPICATs in *met1*, *ddm1*, and *suvh456*. The presence of the Telobox sequence around EPICATs may partially explain the accumulation of H3K27me3 at the heterochromatic regions upon the loss of DNA methylation and H3K9 methylation[43].

Taken together, we conclude that the *A. thaliana* genome harbors hundreds of potential TSSs equipped with functional core

promoter architecture similar to that of regular TSSs. Their activities, however, are suppressed by repressive chromatin restricting their accessibility to transcription machinery.

**Gene body methylation and the suppression of intragenic TSSs.** In *A. thaliana*, about 20% of protein coding genes accumulate CG methylation in their bodies[44]. Moreover, gene body methylation (gbM) is largely conserved across plant species, especially in angiosperms[45], suggesting its functional importance. Although many hypotheses have been proposed regarding the biological functions of gbM, such as suppressing spurious intragenic transcription[25], impeding transcriptional elongation[46], or reducing transcription noise[47], so far its role in plants has been largely elusive[48]. By exploiting the high resolution CAGE-seq data of genome-wide TSSs, we reexamined the relationship between gbM and intragenic transcription initiation in *A. thaliana*. Our data showed that, in wild-type plants, a similar fraction of both body methylated (BM) and non body methylated (non-BM) genes harbored intragenic TSSs, suggesting that the methylation state of gene body is not significantly associated with the occurrence of

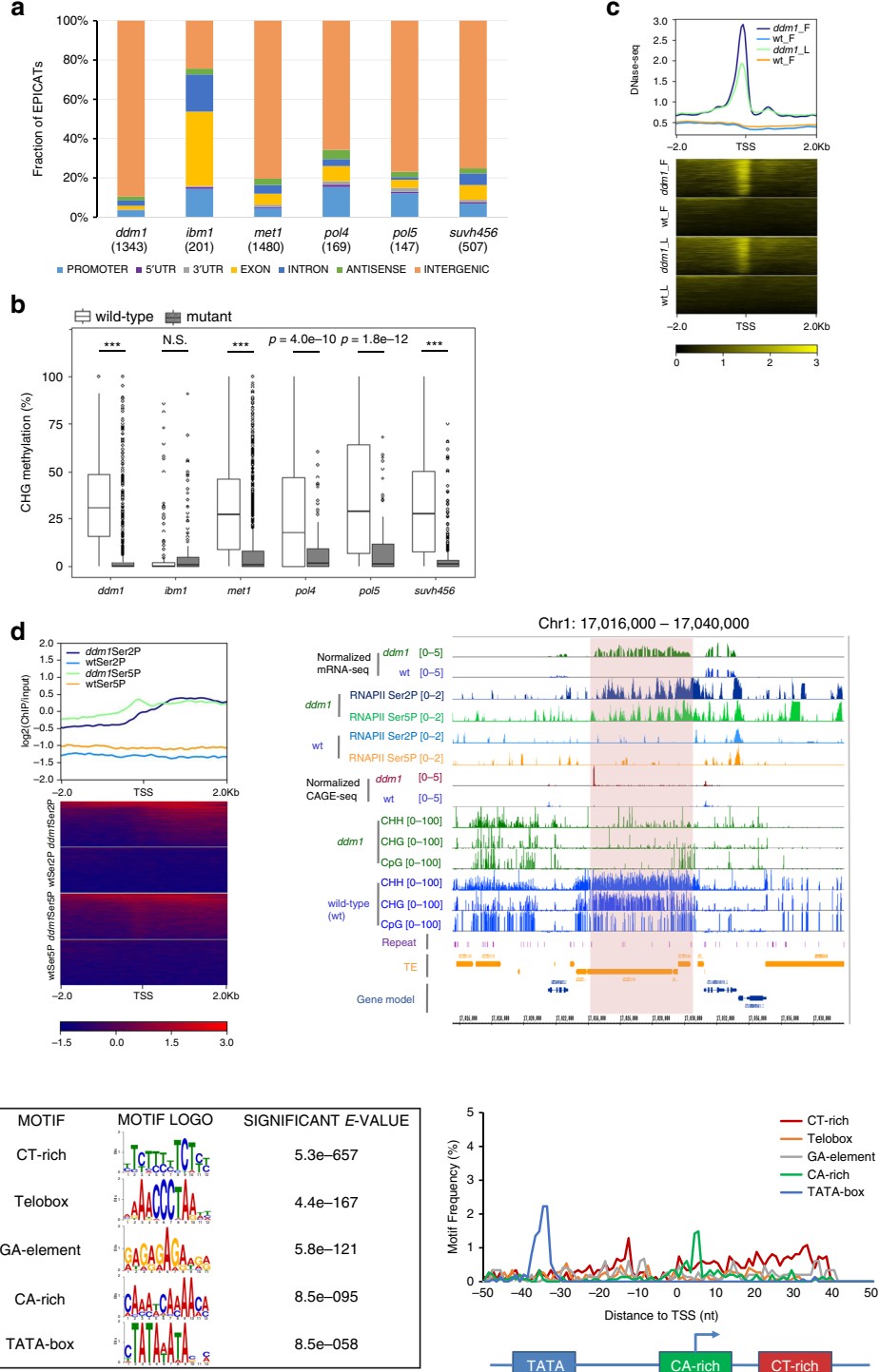

**Fig. 3 Features of the EPICATs activated in epigenetic mutants. a** Genome-wide distributions of EPICATs. Genomic features were ordered as described in Fig. 1a. The number of the EPICATs activated in each mutant is given in the parentheses. **b** Change of CHG methylation, a hallmark of repressive chromatin, at EPICATs in epigenetic mutants. Methylation levels were calculated for 101 bp regions centering around the dominant CTSSs of the EPICATs. ***$p < 2.2e$-16; N.S.: Not Significant ($p > 0.01$), given by paired two-sided Student $t$-test. **c** Increase of chromatin accessibility at the EPICATs in the *ddm1* mutant compared to wild-type (wt) plants, measured by DNase-seq data in flower (F) and leaf (L) tissues. Signals were calculated for non-overlapping 50 bp windows in the regions of ±2 Kb centering around the EPICATs, aligned by their dominant CTSSs (indicated by TSS), and then sorted by the average value of each row. **d** Ectopic recruitment of transcription machinery, represented by RNAPII phosphorylated at Serine 2 (Ser2P) and Serine 5 (Ser5P), to the EPICATs in *ddm1*, at the genome-wide scale (heatmap, left) and a representative locus (indicated by orange windows in the browser tracks, right). ChIP-seq signals in the heatmap were calculated for non-overlapping 50 bp windows in the regions of ±2 Kb centering around the EPICATs, aligned by their dominant CTSSs (indicated by TSS), and then sorted by the average value of each row. **e** Top 5 enriched motifs (left) and their spatial arrangement (right) at the EPICATs in *met1*. Significant $E$-value and the motif logos were given by de novo motif analysis. Distances were calculated between the midpoints of motif instances ($p$-value ≤ 1e-04) and the dominant CTSSs of the EPICATs, and the average profile of all motif instances is shown.

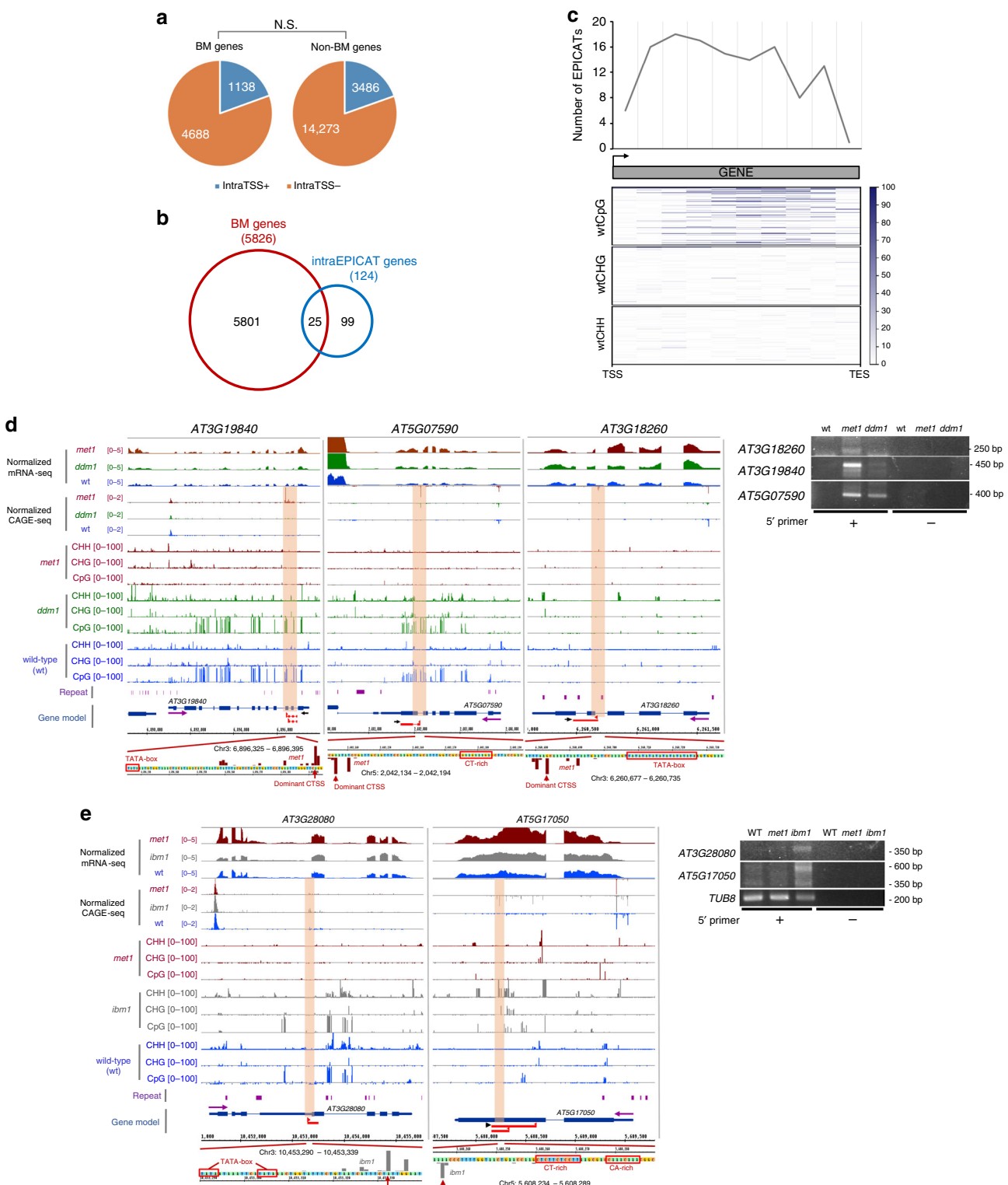

intragenic TSSs (Fig. 4a). Moreover, only a few BM genes activated intragenic EPICATs when gbM was strongly lost in *met1* background (Fig. 4b, Supplementary Fig. 9a), meanwhile intragenic EPICATs could be activated at some loci without gbM (Fig. 4d). These evidences, which are consistent with the conclusions of a previous study[48], suggested that gbM alone is dispensable for suppressing intragenic transcription at a global scale in *A. thaliana* (Supplementary Fig. 9b). Although some BM genes harbored intragenic EPICATs in *met1* (Fig. 4c, d), at this time, we

do not know if this is a direct or indirect effect of *met1* mutant. Future testing using targeted demethylation could help resolve if BM is causal at these loci.

The intragenic EPICATs in *met1* may correspond to 5′-end capped products of post-transcriptional processing of mature mRNAs generated at the associated gene loci, a mechanism well-described in mammals[32,33]. Although we did not rule out this possibility, our data provided evidences supporting that some of these EPICATs are genuine TSSs. First, these loci exhibited a

**Fig. 4 Gene body methylation (gbM) is not significant for suppressing spurious intragenic transcription in *Arabidopsis*. a** Fractions of body methylated (BM) and non body methylated (Non-BM) genes harboring intragenic TSSs (IntraTSS+) or not (IntraTSS-) in wild-type *A. thaliana*. N.S.: not significant, Fisher's exact test. **b** Overlap between body methylated genes (BM genes, red) and genes containing intragenic EPICATs activated in *met1* (intraEPICAT genes, blue). **c** Metaplot showing DNA methylation at genes harboring intragenic EPICATs in *met1*. Gene body was divided into 10 equal bins. DNA methylation levels (the heatmaps in lower panel) and the number of intragenic EPICATs (upper panel) located within each bin were then calculated. TAIR10-annotated TSS and transcription direction are denoted by black arrow. TSS/TES: Transcription Start/End Sites. **d** Browser tracks (left) showing the activation of intragenic EPICATs (indicated by orange windows) at *AT3G19840*, *AT5G07590*, and *AT3G18260*, gene loci in the *met1* and *ddm1* backgrounds. The existence of cryptic transcripts initiated from these EPICATs was confirmed by 5′ RACE (right panel). Purple arrows indicate the direction of transcription. Red lines indicate transcripts detected by 5′ RACE. Red arrows indicate the 5′ end of the transcripts. Black arrows indicate gene specific primers used in 5′ RACE. Four independent clones were sequenced for each genotype, and transcript variants detected in at least two clones are shown. Promoter-associated DNA sequences are indicated in red boxes. **e** Browser tracks (left) showing the activation of intragenic EPICATs (indicated by orange windows) at *AT3G28080* and *AT5G17050* gene loci in the *ibm1* and *met1* backgrounds, validated by 5′ RACE experiments (right). Data are presented as described in **d**.

stronger accumulation of RNAPII in *met1* (Supplementary Fig. 9c). Second, only 1/124 genes harboring intragenic EPICATs also had upstream EPICATs (Supplementary Fig. 9d), suggesting that these intragenic EPICATs correspond to independent, de novo transcribed mRNAs. Third, promoter-associated DNA sequences were also present at some of these intragenic loci (Fig. 4d).

Besides *met1*, *ibm1* also activated a comparable number of intragenic EPICATs (Supplementary Fig. 6d, Supplementary Data 4). However, it is unlikely that they are directly regulated by the activity of IBM1 (Fig. 3b, Supplementary Fig. 10a). On the other hand, although the expression of *IBM1* is significantly reduced in *met1* background[49], the intragenic EPICATs activated in *ibm1* and *met1* were largely un-overlapped (Supplementary Fig. 10b). Moreover, the accumulation of RNAPII at these loci was not significantly affected in *ibm1* background (Supplementary Fig. 10c), suggesting that intragenic EPICATs in *ibm1* and *met1* are regulated differently. Given that none of the associated genes simultaneously harbored upstream EPICATs, and that promoter-associated DNA sequences were present at some of these intragenic targets (Fig. 4e), we speculate that some of them are genuine TSSs, while some others could be derived from post-transcriptionally processed mRNAs.

**RNAPII and PolIV exclusively bind to RdDM-regulated EPI-CATs.** It has been reported that, although PolIV-dependent RNAs (P4RNAs) feature PolII-like TSSs, PolIV and PolII target distinct genomic territories[50]. Our data, however, showed that 24 nt siRNAs were highly enriched at genomic loci harboring the EPICATs activated in the mutants of the RdDM pathway's components, such as *pol4* and *pol5* (Supplementary Fig. 7c). The biogenesis of these 24 nt siRNAs was indeed dependent on PolIV, which is responsible for the transcription of P4RNAs initiated from the corresponding EPICATs (Supplementary Fig. 11a–c). Moreover, in *pol4* and *pol5* backgrounds, RNAPII was highly recruited to these loci (Supplementary Fig. 7e). These evidences suggest that, genomic regions harboring the EPICATs regulated by the RdDM pathway likely possess distinct features compared to those of its regular targets, which allow PolII and PolIV exclusively function at these loci (Supplementary Fig. 11d).

**TEs are a major supplier of cryptic TSSs in *Arabidopsis*.** The existence of a large number of cryptic TSSs within a small and compact genome, like that of *A. thaliana*, has raised important questions regarding their origin. Investigations involving mammalian genomes have shown that TEs are a major genetic element that can be exapted as TSSs in the host genomes[51,52]. Although less prevalent, several lines of study have demonstrated a similar function of TEs in plant genomes[53,54]. Together with the evidence

that EPICATs are mainly located at intergenic regions decorated with repressive chromatin modifications (Fig. 3a, Supplementary Fig. 7a, b), we speculated that many cryptic TSSs in the *A. thaliana* genome may have originated from TEs. The data indicated that TEs contribute to up to 65% of the EPICATs activated in the mutant backgrounds (Supplementary Fig. 12a). Additionally, hundreds of TEs harboring active TSSs were identified in wild-type background (Fig. 5a, Supplementary Data 5). TEs, therefore, may serve as a reservoir of potential functional TSSs in *A. thaliana*, similar to their role in animal genomes.

There are numerous types of TEs with different origins and mobility strategies[1,2] which greatly affect their abilities to induce genetic variations to the host genomes. Therefore, the TSS-encoding potential of each TE family in the *A. thaliana* genome was examined. Although EPICATs were associated with various TE families (Fig. 5a), compared to the genome-wide average, LTR/Gypsy members were enriched among TEs harboring the EPICATs in *ddm1* and *met1* ($p = 2.0e-52$ and $6.0e-49$, respectively, Hypergeometric test), while members of the LTR/Copia family were highly represented among the TE targets of *ddm1* and *suvh456* ($p = 8.0e-10$ and $2.0e-31$, respectively, Hypergeometric test). In addition, the DNA/En-Spm family was highly associated with the EPICATs in *met1*, *ddm1*, and *suvh456* (Fig. 5a, $p < 1.6e-16$ for all, Hypergeometric test). Due to the minor numbers of TE instances associated with the EPICATs in *ibm1*, *pol4*, and *pol5*, they were skipped from enrichment analysis. The data suggest that both retro- and DNA transposons are genetic suppliers of cryptic TSSs in the *A. thaliana* genome.

Since *ddm1* affected the largest number of TEs harboring EPICATs, and these elements largely overlapped with TEs activated in other mutants (Fig. 5a, Supplementary Fig. 12b), we examined if they possess any specific features that facilitate their ectopic activation in *ddm1* background. Compared to their counterparts, which either contain active TSSs in wild-type plants or do not harbor any EPICATs, TEs harboring EPICATs were more highly methylated in both CG and non-CG contexts (Fig. 5b). They were also substantially longer (Fig. 5b), suggesting that these TEs are likely younger insertions that still maintain intact structures with transcription and transposition capacities, that may be a trigger for greater accumulation of DNA methylation and other repressive modifications at the associated loci. Analysis of the core promoter motifs identified at the *ddm1*-activated EPICATs (Supplementary Fig. 8) showed that they were more prevalent among EPICAT-harboring TEs (Fig. 5c). However, there were still hundreds to thousands of inactive TEs associated with these motifs (Supplementary Fig. 12c). As a case study, the genetic structure associated with the EPICATs located in the LTR regions of the Gypsy TEs was investigated in a more detail. This was because the LTR/Gypsy family contributed a large number of elements harboring the EPICATs in *ddm1* and *met1*

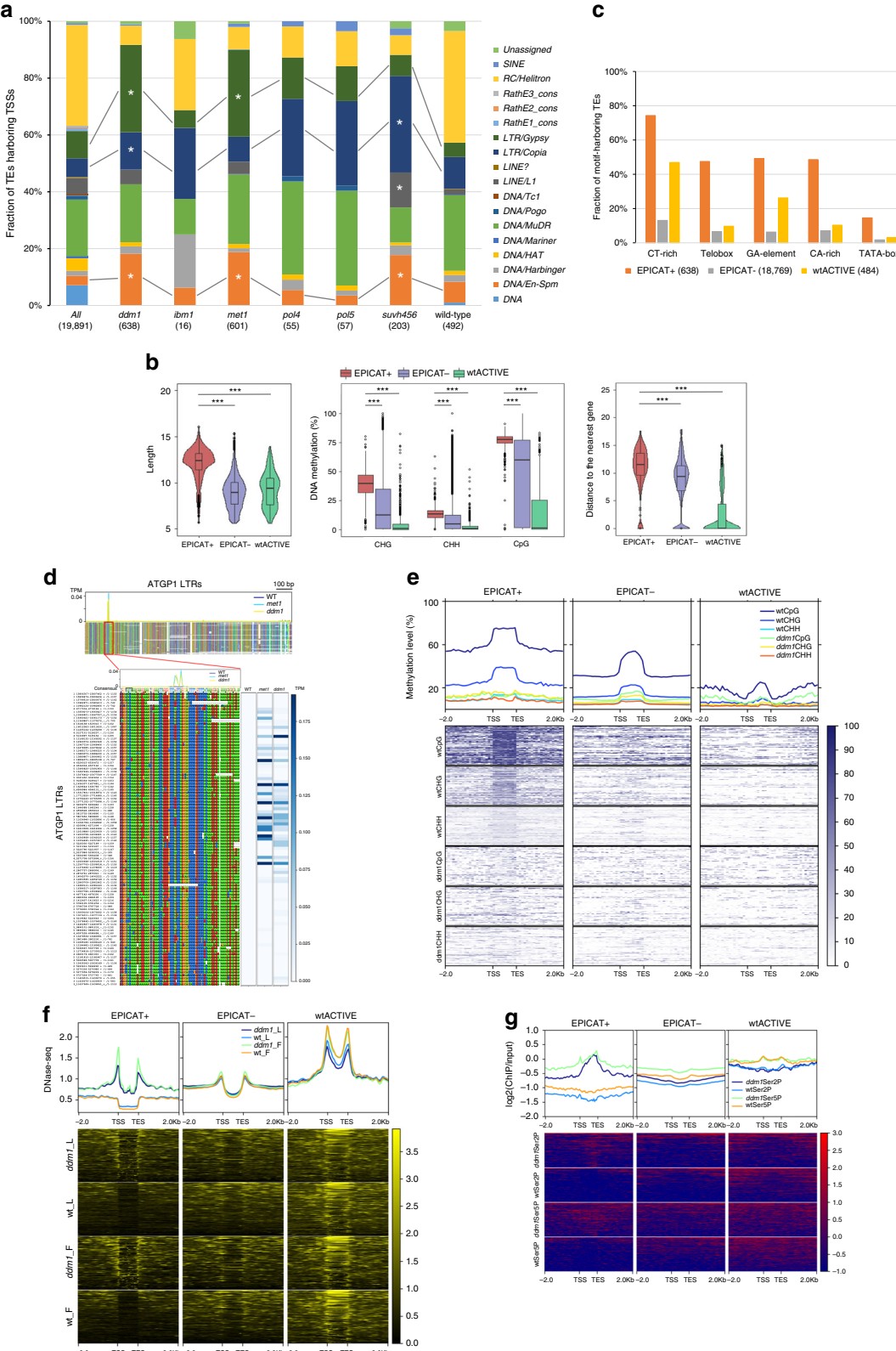

(Fig. 5a), and its members still maintain transcription/transposition potential in the *Arabidopsis* genome[55]. Although LTR sequences surrounding the CAGE-seq peaks were largely diverged between and within Gypsy sub-families, they commonly shared putative TATA-box and TSS-associated YR motifs (Fig. 5d, Supplementary Fig. 12d). However, the conservation of sequences/motifs surrounding the LTR-encoded TSSs could not

fully explain their activation in the mutant backgrounds. Moreover, although a significant loss of repressive modifications (e.g., DNA methylation) was observed at many TEs regardless of their association with the EPICATs in *ddm1* (Fig. 5e), only EPICAT-harboring elements became highly accessible in the mutant, especially at their two ends (Fig. 5f). Concomitantly, RNAPII was highly recruited to these loci, together with an

**Fig. 5 TEs are a major genetic supplier of cryptic TSSs in the *A. thaliana* genome. a** Transposon families harboring the EPICATs. Shown are total numbers of TEs. All, wild-type: TEs in the *A. thaliana* genome and TEs harboring active TSSs in wild-type plants, respectively. *TE families enriched compared to the genome-wide average ($p < 3e-04$, Hypergeometric test). **b** Violin and/or boxplots showing the Length (left), DNA methylation (middle), and Distance to the nearest gene (right), of TEs harboring (EPICAT+), not harboring (EPICAT−) the EPICATs in *ddm1*, and harboring active TSSs in wild-type plants (wtACTIVE). ***$p < 2.2e-16$, two-sided Mann–Whitney test. **c** Fractions of TEs harboring consensus DNA motifs found at the *ddm1*-activated EPICATs. **d** Sequence alignment of representative LTRs of TEs belonging to the Gypsy ATGP1 sub-family. LTR sequences were aligned based on relative positions of cryptic TSSs detected by CAGE-seq. A, C, G, T nucleotides were colored in green, yellow, red, and blue. Putative TATA-box and YR-motifs are indicated by black box and red line, respectively. Relative positions and strand information of each LTRs in the genome are indicated (bottom left). Heatmaps of the normalized CAGE-seq data mapped to the LTRs using one replicate of each mutants are shown on the right side of the bottom panel. **e** DNA methylation of TEs in the wild-type (wt) and *ddm1* backgrounds. TE lengths were normalized to 1 Kb and aligned by their two ends (indicated by TSS and TES). Methylation levels were calculated for non-overlapping 100 bp windows within TE body and surrounding region of ±2 Kb, and sorted by the average value of each row. **f** Chromatin accessibility at TEs in *ddm1* and wild-type (wt) plants, given by DNase-seq data in flower (F) and leaf (L) tissues. Signals were calculated and presented as described in **e**, with the window size of 50 bp. **g** Ectopic recruitment of RNAPII Ser2P and Ser5P to TEs in the wild-type (wt) and *ddm1* plants. Signals were calculated and presented as described in **e**, with the window size of 50 bp.

increased production of the associated transcripts (Fig. 5g, Supplementary Fig. 12e). These data suggest that, in addition to the presence of core promoter sequences, factors regulating chromatin environment are required for RNAPII recruitment and the ectopic activation of TE-encoded EPICATs.

**Regulatory impact of transcription from cryptic TSSs.** In mammals, TE sequences frequently act as alternative promoters to regulate development-associated gene expression programs[51,52]. While the contribution of TEs to plant transcriptomes has been much less clear[56], this evidence suggests that regulatory elements supplied by TEs can be co-opted for transcriptional regulation in plant genomes[28]. Using the EPICATs activated in *met1* as a proxy, we therefore investigated the potential alteration in the *A. thaliana* transcriptome induced by cryptic TSSs. About ~80% of the EPICATs in *met1* were associated with the transcripts assembled from mRNA-seq data (Supplementary Fig. 13a, Supplementary Data 6, see the "Methods" section for details). Moreover, the expression of EPICATs was positively correlated with that of the assembled gene units (Supplementary Fig. 13a–c). 73% of the transcripts associated with *met1*-activated EPICATs had more than one exons, of which 112 (~9%) shared splicing junctions with 75 reference gene units (Fig. 6a). Surprisingly, about half (50/112) of these spliced transcripts possessed at least one active TSS in wild-type background, suggesting that their regular transcription, and consequently downstream functions, can potentially be affected by the ectopic activation of EPICATs. We selected and experimentally confirmed the production of novel cryptic fusion transcripts at some of these loci in *met1* and/or *ddm1* backgrounds, which include *SQN (AT2G15790)*, a gene critical for vegetative shoot maturation[57], *COQ3 (AT2G30920)*, a gene encoding a mitochondria-localized methyltransferase important for ubiquinone biosynthesis and embryo development[58,59], and a gene of unknown function (*AT2G16050*) (Fig. 6b, c, Supplementary Fig. 14a, b). To complement the CAGE-seq data, transcripts with significant alteration in promoter usage were analyzed using mRNA-seq data (see Methods section for details). Of the resulting transcripts, 10 were found associated with *met1*-activated EPICATs at three gene loci (Supplementary Data 7). We also experimentally confirmed the production of a read-through fusion transcript from the annotated TSS at the *AT5G28442* gene locus, which harbored an EPICAT in *met1* and *ddm1* backgrounds (Supplementary Fig. 14a, b).

Although it has been suggested that repressive chromatin associated with TE insertions potentially imposes negative impacts on the transcription of nearby genes[13,14], direct consequences of TE-encoded TSS activation on the surrounding transcriptional environment remain obscure. Inspection of the loci producing cryptic fusion transcripts revealed that some of them concurrently exhibited reduced transcription from their regular TSSs in the mutant backgrounds (Fig. 6b, Supplementary Fig. 14a). This suggests that, the activation of EPICATs may also quantitatively affect the transcription from nearby regular TSSs. Therefore, wild-type active TSSs located in the vicinity (up to 3 kb) of EPICATs were examined to see how their expression is altered upon EPICAT activation. While some showed increased expression, the majority were not significantly affected (Fig. 6d, e). Nevertheless, there were groups of TSSs whose expressions were significantly suppressed in concomitant with the activation of nearby EPICATs (Fig. 6e, Supplementary Data 8). Of the gene loci associated with the TSSs suppressed in *met1*, five were selected for validation by qPCR. Except *AT5G28442*, which could not be amplified, significant decreases in the expression at three out of the four loci in *met1* and *ddm1* were confirmed, which is consistent with the observation from the CAGE-seq data (Fig. 6f, Supplementary Fig. 14c). These include *AT1G23935*, *SUS5 (AT5G37180)*, and *PRB1 (AT2G14580)*, a gene involved in response to abiotic stress in *Arabidopsis*[60].

Taken together, these data demonstrate that the activation of cryptic TSSs has critical impacts on the transcriptome of *A. thaliana*, both qualitatively and quantitatively.

## Discussion

To understand how transcription initiation in plants is epigenetically regulated, we have generated a comprehensive maps of TSSs in various epigenetic mutants of *A. thaliana* using CAGE-seq. Compared to mammals, epigenetic mechanisms regulating transcription initiation in plants are much less clear, mainly due to a lack of suitable resources which allow the investigation of the alteration of transcription initiation under different conditions[25,26,56]. This study, therefore, provides valuable reference data for research communities to enlighten the impact of epigenetic regulation on transcription initiation landscapes in plants.

Our study showed that, in epigenetic mutant backgrounds, thousands of cryptic TSSs are activated, in which the mutant of maintenance DNA methylation *met1* regulates the largest number of targets (Fig. 2a). A large number of cryptic TSSs reside in TE sequences, which are dominantly contributed by members of the LTR/Gypsy, LTR/Copia, and DNA/En-Spm families (Fig. 5a). Interestingly, there is a clear difference in DNA methylation between TEs with and without EPICATs, where the former accumulate higher DNA methylation (Fig. 5b, e). This suggests that the DNA methylation of TEs could be largely influenced by their potential to initiate transcription. On the other hand, the analysis of LTR sequences indicated that the conservation of core promoter elements alone is not sufficient for transcription

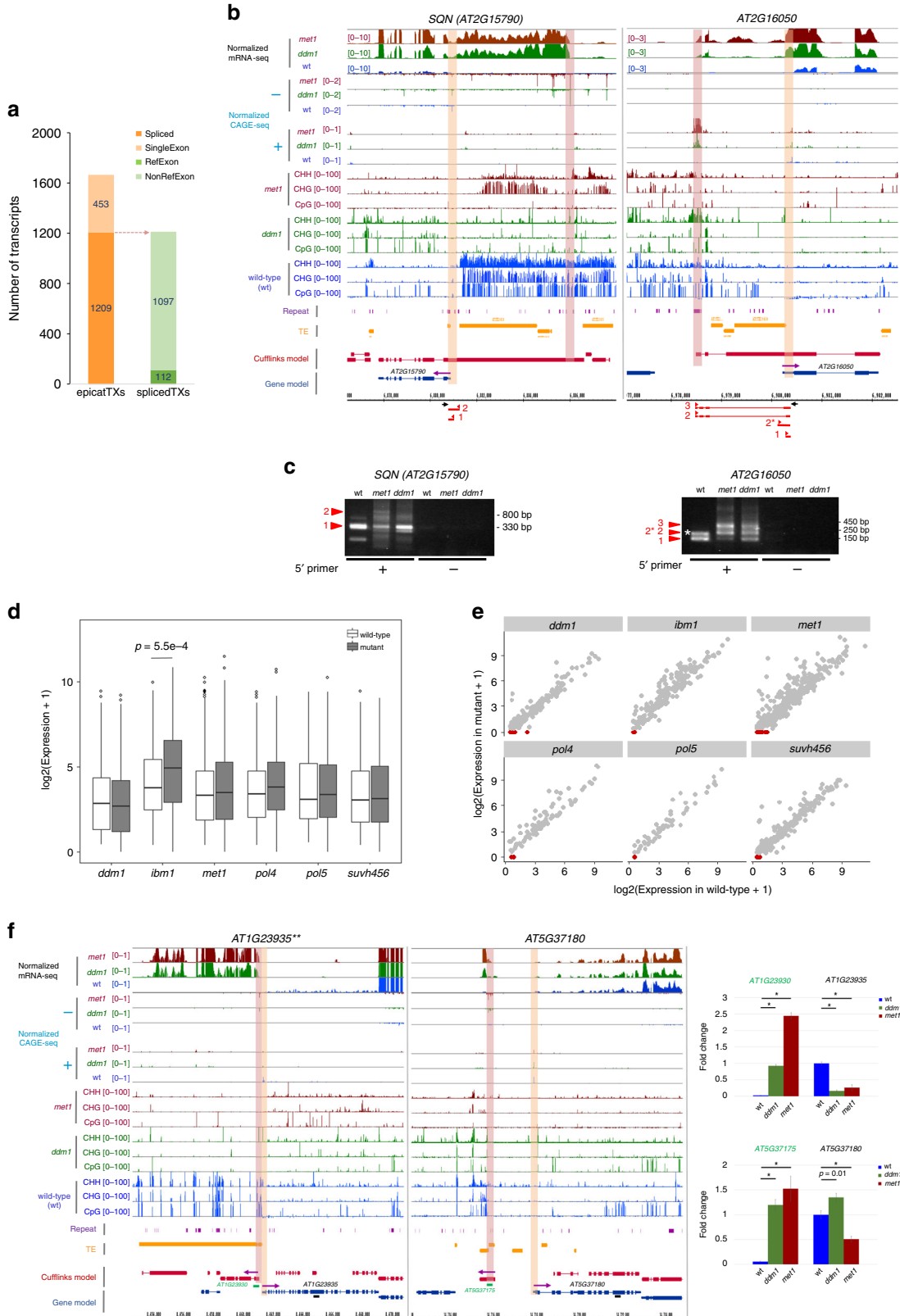

initiation (Fig. 5d, Supplementary Fig. 12d) as their transcription levels are largely varied, even among LTRs with nearly identical sequences. The ability of TE-encoded TSSs to initiate transcription may, therefore, also be dependent on their relative positions within TEs (e.g., whether they are located at the 5′- or 3′-end of the TEs), and/or local chromatin environments, such as higher-order chromatin conformation and long-range enhancer interactions[61].

In mammals, the loss of gene-body DNA methylation caused by *DNMT3b* knockout triggers spurious RNAPII recruitment and cryptic transcription initiation from intragenic regions[25]. The analysis of intragenic TSSs in the present study showed that a

**Fig. 6 Impacts of spurious transcription from cryptic TSSs on the *A. thaliana* transcriptome. a** Classification of transcripts associated with the *met1*-activated EPICATs (epicatTXs). Spliced transcripts (splicedTXs) were further divided into (RefExon) and (NonRefExon) if they share splice junctions with reference gene units or not, respectively. Shown are total numbers of transcripts. **b** Browser tracks showing the alteration of transcription at *SQN* (*AT2G15790*) and *AT2G16050* gene loci upon the activation of nearby EPICATs in the *met1* and *ddm1* backgrounds. Positions of regular and cryptic TSSs are indicated by green and orange windows, respectively. Purple arrows indicate the direction of transcription. **c** Detection of cryptic fusion transcripts at the *SQN* and *AT2G16050* loci in the *met1* and *ddm1* backgrounds by 5′ RACE. The detected transcripts are denoted by red lines, their 5′ ends are indicated by red arrows, and gene specific primers used in 5′ RACE are indicated by black arrows in Fig. 6b. Four independent clones were sequenced for each genotype, and transcript variants detected in at least two clones are shown. **d** The expressions of regular TSSs located in the vicinity of the EPICATs in the wild-type and mutant plants. Only TSSs active in the wild-type plants and located within promoters or 5′ UTRs of annotated genes were considered. *p*-values were calculated by paired two-sided Student's *t*-test. $p \leq 0.01$ are shown. **e** Similar to **d**, but presented in scatter plots. Regular TSSs significantly suppressed in the mutants (expression in mutant = 0) are marked in red. **f** The attenuation of transcription at the *AT1G23935* and *AT5G37180* loci upon the activation of nearby EPICATs in the *met1* and *ddm1* backgrounds, shown in browser tracks and validated qPCR. The mean values of the wild-type samples were set as 1, and relative fold changes in each sample were shown in bar graphs (mean $+/-$ S.E.M.; $n = 7$–8 samples for each genotype). Black and green bars indicate the positions used for qPCR analysis. Regular TSSs and nearby EPICATs are indicated by green and orange windows in the browser tracks, respectively. Purple arrows indicate the direction of transcription. *$p < 0.0005$, given by paired Student *t*-test. **Identified using three *met1* CAGE-seq replicates.

complete loss of gbM in the *met1* mutant does not profoundly activate intragenic transcription in the *Arabidopsis* genome (Fig. 3a, 4, Supplementary Fig. 9a, b). Recruitment of DNMT3b to genic regions in mammals is dependent on histone H3K36 methylation[62]. In yeast, H3K36 methylation (H3K36me) mediated by SET2 suppresses cryptic intragenic transcription initiation[63]. In plants, however, concurrent loss of both gbM and H3K36me3 does not show significant difference in transcription between (BM) and unmethylated (UM) loci[48]. On the other hand, regulation of cryptic transcription from intronic heterochromatin by the RdDM pathway[64], and the suppression of intragenic antisense transcripts by histone H1 and DNA methylation[65] have also recently been reported. These results suggest that plants may employ additional layers of epigenetic regulation to prevent spurious transcription initiation, especially in intragenic regions.

The activation of spurious transcription from cryptic TSSs would inevitably alter transcription from nearby regular TSSs (Fig. 6, Supplementary Fig. 14). The data showed that such alteration may occur in several different scenarios. First, an activated cryptic TSS located upstream may function as the major initiation site facilitating the formation of a read-through transcript, which can suppress transcription from a downstream regular TSS, as observed at *AT2G16050* and *SQN* loci (Fig. 6b). This regulatory effect is likely facilitated by a less understood mechanism known as transcriptional interference[66,67]. Secondly, the activation of a cryptic TSS located downstream may attenuate transcription initiated from an upstream regular TSS and trigger the production of spurious transcripts, as observed at *AT2G14580*, *AT2G15042*, and *AT5G28442* loci (Supplementary Fig. 14). Thirdly, when cryptic and regular TSSs are situated close to each other, but in divergent directions, transcription from the regular TSS may also be suppressed (Fig. 6f). Such repressive impacts could be facilitated by competitive binding to regulatory sequences of transcription initiation complexes associated with the two TSSs[66], or by the mechanism suppressing transcription from divergent promoters[68], or by the lack of a mechanism facilitating bi-directional transcription in plants[20] compared to mammals[37].

Whether the epigenetic regulation of cryptic TSSs brings any potential developmental and/or adaptive advantages or disadvantages to a plant species is of great interest in plant research. As epigenetic information is relatively flexible and can be reprogrammed according to environmental stimuli, the mechanisms described here may provide plants with a fast and efficient mean for tuning, or even inverting the polarity of regulatory inputs on, gene expression. In addition, potential activation and co-option of cryptic TSSs can provide alternative promoters to the existing transcription units, as observed at

*AT2G16050* and *COQ3* loci (Fig. 6b, c, Supplementary Fig. 14a, b), which may help plants customize gene functions during development[51,52]. Such events can also create opportunities for plants to innovate their transcriptome in response to environmental changes. However, the mis-control of cryptic TSSs encoded in TEs may trigger developmental abnormality in plants[11,69]. In addition, modulating 3′ and/or 5′ UTRs of a transcript without changing its coding potential can critically affect its function in response to pathogen attacks in *Arabidopsis*[70]. Epigenetic suppression of a cryptic TSS at the 5′ UTR of the LRR gene *AT2G15042* (Supplementary Fig. 14a) may, therefore, help maintain the proper response of *Arabidopsis* to viral infection[71]. Importantly, activation of the cryptic TSS upstream of *SQN* (*AT2G15790*), a gene important for vegetative shoot maturation in *Arabidopsis*[57], leads to ectopic production of aberrant transcripts and a decreased accumulation of the normal one (Fig. 6b). Although the impacts of such transcriptional attenuation on plant development are to be confirmed, it has been shown in *A. thaliana* that, light-induced regulation of alternative promoters could generate proteins with differential localizations from the same genes, which help alleviate the impact of changing light conditions on the plant[72]. Our data, therefore, demonstrate that the epigenetic regulation of cryptic TSSs would profoundly and critically affect proper responses of plant species to ever changing environmental conditions. Additionally, as many protein coding genes in *A. thaliana* possess multiple active upstream as well as intragenic TSSs, it would be interesting to investigate whether cryptic TSSs are still in the process of being co-opted to become functional in the *Arabidopsis* genome.

## Methods

**Plant materials**. *ddm1-1*, *met1-3*, *ibm1-4*, *ibm2-2*, and *edm2-9* mutants have been described previously[16,73–75]. *suvh456* and *nrpe1-7* seeds were kindly provided by Dr. Kakutani and Dr. Kanno, respectively. The T-DNA insertion line of *nrpd1a-3* (SALK_128428) was obtained from the Arabidopsis Biological Resource Center (https://abrc.osu.edu). All the mutants are in Columbia (Col) background. The second generation of homozygous *met1*, *ddm1*, *ibm1*, *ibm2*, and *edm2* were used for the RNA experiments described below. *nrpd1a*, *nrpe1*, and *suvh456* were maintained as homozygous for at least three generations before the experiments. The seeds were germinated and grown on 1/2 Murashige and Skoog (MS) plate under long-day conditions (16-h light; 8-h dark) at 22 °C.

**RNA extraction and CAGE**. For CAGE analysis, 10-to-12-day-old whole seedlings of wild-type Col and mutant plants were pooled for RNA extraction. Total RNA was extracted using RNAiso (TAKARA), and DNA was digested with TURBO DNase (Thermo Fisher Scientific), followed by purification by RNeasy Plant Minikit (QIAGEN). Four technical replicates of WT Col and *met1*, and two technical replicates of other samples were prepared for CAGE. Single end 75bp CAGE libraries were prepared and sequenced in DNAFORM (Yokohama, Japan). RNA quality was assessed by Bioanalyzer (Agilent) to ensure that the RIN (RNA integrity number) was over 7.0, and A260/280 and 260/230 ratios were over 1.7.

**CAGE sequencing data analysis**. The CAGE sequencing (CAGE-seq) data were processed as follows: sequencing reads were trimmed using Trimmomatic (v0.30)[76] with the following parameters: HEADCROP:1, TRAILING:20, to remove nonspecific guanines[38] and low quality bases at the read ends. These were then mapped to the *Arabidopsis* Col reference genome by HISAT2 (v2.0.0-beta)[77], allowing up to ten alignments for a single read. Due to low mapping coverage, met1.4 replicate was excluded from further analysis. met1.3 was also discarded due to its low correlations with two other replicates (met1.1 and met1.2). Then, uniquely mapped reads were used to identify TSSs at a single base resolution (CTSSs) by CAGEr (v1.20.0)[78] with the following parameters: sequencingQualityThreshold = 20, mappingQualityThreshold = 20. After being normalized to Tags Per Million (TPM), CTSSs in each sample were grouped into tag clusters by the paraclu method, with threshold = 0.1, nrPassThreshold = 2, removeSingletons = TRUE, keepSingletonAbove = 0.3, minStability = 2, max-Length = 100. Finally, tag clusters from individual samples were merged into a common set of consensus tag clusters by the aggregateTagCluster function, with threshold = 0.3, $q$Low = NULL, $q$UP = NULL, maxDist = 100, exclude-SignalBelowThreshold = TRUE. Each consensus tag cluster was then considered a single reliable TSS, represented by its dominant CTSS, to distinguish from the TSSs annotated by TAIR10. Promoter width was defined by the distance between the 10th ($q$Low = 0.1) and 90th ($q$Up = 0.9) quantiles of the cummulative distribution of CAGE signal along each tag cluster, as described in ref. [78]. Raw tag counts were used to identify differentially expressed TSSs in the mutants compared to wild-type plants by DESeq2 (v1.22.2)[79], with significance cut-off threshold $p$adj ≤ 0.1.

**Annotating TSSs identified by CAGE-seq**. TAIR10 genome annotations of 19,891 TEs and 27,600 protein coding genes and non coding RNAs in *A. thaliana* were obtained from ref. [14]. Araport11 version of genome annotations were also downloaded from The Arabidopsis Information Resource (TAIR) (https://www.arabidopsis.org/). Promoters were defined as the regions of 1 kb upstream of the TAIR-annotated TSSs. A TSS identified by CAGE-seq was annotated based on genomic location of its dominant CTSS, in the following order: promoter, 5′ UTR, 3′ UTR, intron, exon, antisense, TE, intergenic.

TSSs identified by PEAT method were obtained from ref. [31]. Then, the nearest distance between the dominant CTSS of each CAGE-seq tag cluster and the mode locations of PEAT TSSs in the same direction was calculated. PEAT TSSs, which exactly matched with CAGE-seq TSSs (distance = 0 nt), were used as the proxy to estimate interquantile widths for each shape category defined in ref. [31], including NP, broad with peak (BP), and weak peak (WP).

**mRNA sequencing data analysis**. Paired-end mRNA sequencing (mRNA-seq) data were prepared following the method described in ref. [14] and processed as follows: reads were trimmed by Trimmomatic to remove sequencing bias and adapter sequences, then mapped to the *Arabidopsis* Col reference genome by HISAT2, allowing up to ten alignments for a read pair. The featureCounts function in the package Rsubread (v1.14.2)[80] was used to identify the number of read pairs uniquely mapped to genes and TEs.

The outputs of mRNA-seq mapping were also used for transcript assembly as follows: first, transcripts of each individual sample were assembled by Cufflinks (v2.2.1)[81]. Low-expressed transcripts (smaller than the 10th percentile of expression of all the assembled transcripts) were then removed. The remaining transcripts from all samples were merged to create a unified set of transcripts. They were then compared to reference transcripts in TAIR10 by the cuffcompare function to identify splicing patterns. Differential promoter usage was assessed by the cuffdiff function.

To identify assembled transcripts associated with EPICATs, overlap tests were conducted between the transcripts and genomic regions centering around the EPICATs' dominant CTSSs (extended 180 bp into both sides, regarding that a TSS identified by CAGE-seq could be associated with a nearby transcript (Supplementary Fig. 2b)). The results were given in Supplementary Data 6.

**ChIP sequencing data analysis**. ChIP sequencing (ChIP-seq) data of histone modifications, including H3K27me1/3, H3K9me2, H3K36me3, and H3K4me3, in wild-type plants were retrieved from a previous study[82]. Paired-end Chip-seq data of RNAPII in wild-type plants and mutants were prepared as follows: Two-week-old whole seedlings of wild-type Col and *met1* and *ddm1* were fixed in a fixation buffer (10 mM Tris-HCl (pH 7.5), 50 mM NaCl, 0.1 M sucrose, 1% formaldehyde) for 20-min, followed by quenching by 125 mM Glycine. Nuclei isolation was performed as previously described[83]. PolII ChIP was performed for two replicates for each genotype (about 1 g tissue/IP) by SimpleChIP Plus Kit (Cell Signaling Technology) according to the manufacturer's instructions. Anti-RNA polymerase II CTD repeat YSPTSPS (phospho S2) (Abcam ab5095) and Anti-RNA polymerase II CTD repeat YSPTSPS (phospho S5) (Abcam ab5408) antibodies were used for IPs (4 μg/IP). Precipitated DNA samples were sequenced by Hiseq 4000 in the 150 bp paired-end mode in OIST SQC. Due to the large overlap between two reads, only one read (read 1) in each pair was used for downstream analysis. Reads were

trimmed to remove sequencing bias and adapter sequences using Trimmomatic, then mapped to the *Arabidopsis* Col reference genome by Bowtie (v1.0.0)[84]. Reads mapped to an identical position were collapsed into a single read, and only the best alignment was kept for a read mapped to multiple locations. Mapping results were given in Supplementary Data 9.

ChIP-seq data of PolIV (NRPD1) and the list of NRPD1 binding loci were obtained from ref. [85]. Genomic locations of NRPD1 binding loci were then converted from TAIR8 to TAIR9 coordinates using the *update_coordinates.pl* script provided by TAIR. ChIP-seq data of RNAPII in *pol4* and corresponding wild-type plants were obtained from ref. [50]. These data were processed as described above. Preprocessed RNAPII Ser5P ChIP-seq data (in bigwig format) in *pol5* were downloaded from ref. [64] and directly used for visualization.

**Bisulfite sequencing data analysis**. Whole-genome bisulfite sequencing (WGBS) MethylC-Seq data of wild-type plants and epigenetic mutants were retrieved from ref. [9]. High quality reads ($q ≥ 28$), trimmed to remove adapter effects and sequencing bias, were mapped to the *Arabidopsis* Col reference genome using Bismark (v0.12.1)[86] allowing up to two mismatches. Bases covered by fewer than 3 reads were excluded, and only uniquely mapped reads were used for further analysis. Methylation levels were calculated using MethylKit (v0.5.7)[87]. The list of BM, intermediate methylated (IM), and unmethylated (UM) genes were obtained from ref. [44]. To exclude the potential impacts of non-CG methylation on the activation of intragenic EPICATs, only *met1*-activated intragenic EPICATs with low (less than 10%) CHG methylation in the 101 bp regions centering around their dominant CTSSs were examined (Supplementary Data 4).

**Small RNA sequencing data analysis**. Sequencing data of 24 nt small interference RNAs (siRNAs) in wild-type and *nrpd1* mutant plants were obtained from ref. [85] and trimmed by TrimGalore (v0.4.5)[88] with Cutadapt (v1.8.3)[89], using the following parameters: stringency:4, quality:20, length:15, max_length:30. PolIV-dependent small RNAs (P4RNAs) longer than 27 nt in *dcl2/3/4* and corresponding wild-type plants were obtained from ref. [50] and trimmed by Trimmomatic. These data were then mapped to the *Arabidopsis* Col reference genome by Bowtie (v1.0.0), allowing up to two mismatches. Only uniquely mapped reads were used for further analysis.

**Sequence motif analysis**. De novo motif analysis and search of motif instances were conducted using MEME suite (v4.11.2) with default parameters[90].

**Gypsy LTR analysis**. Gypsy family sequences were retrieved from the TAIR database and aligned to obtain the full-length sequence for each family. LTR regions were then determined by comparing 5′ and 3′ ends of TE sequences and also checked by LTR_FINDER (v1.0.2)[91]. Several copies from each family were used to obtain consensus sequences of LTRs (Supplementary Data 10). Consensus sequences of Gypsy LTRs were used to search for LTR sequences in the *Arabidopsis* genome (TAIR10) using BLAST (v2.0)[92]. BLAST hits shorter than 100 bp were discarded. LTR sequences were then aligned using ClustalW (v2.1)[93], and edited using Jalview (v2.11.0)[94].

**Data visualization**. Figures were created using deepTools (v3.3.0)[95], Integrated Genome Browser (IGB) (v9.1.2)[96] with the Araport11 version of genome annotations, Excel, and the R package ggplot2 (v2.3.1)[97]. DNA methylation files were firstly converted from bedGraph into bigWig format by the bedGraphToBigWig function (http://hgdownload.soe.ucsc.edu/admin/exe/linux.x86_64/), then used to generate heatmap and metaplot figures using deepTools. mRNA-seq data were normalized to reads per million (RPM), and a single replicate was used to create IGB track. ChIP-seq signals were normalized to log2(ChIP/input), and a single replicate of RNAPII (both Ser5P and Ser2P) were used for visualization in IGB. Small RNA sequencing data and RNAPII ChIP-seq data with no input samples were normalized to counts per million (CPM).

**5′-RACE and quantitative PCR**. 5′-RACE was performed by SMARTer RACE kit (TAKARA) according to the manufacturer's instructions. Quantitative PCR (qPCR) was performed following the method described in ref. [75]. All primers used in this study are listed in Supplementary Data 11.

**Reporting summary**. Further information on research design is available in the Nature Research Reporting Summary linked to this article.

## Data availability
Sequencing data have been deposited to the DDBJ Sequence Read Archive under the accession codes DRA009134 and DRA009847. Processed CAGE-seq data are also accessible via the following web link: https://plantepigenetics.oist.jp/. The source data underlying Figs. 1b, 2c, 3b, 4d–e, 5b, and 6c, d, f and Supplementary Figs. 6b–c, 7d, and 14b–c are provided as a Source Data file.

## Code availability

In-house R codes and bash scripts customized for analyzing data are available from the authors upon request.

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

## Acknowledgements

This work was supported by JSPS KAKENHI Grant Number 19K06619 to H.S., and by Okinawa Institute of Science and Technology Graduate University. We thank the Arabidopsis Biological Resource Center and the Salk Institute Genomic Analysis Laboratory for providing *Arabidopsis* T-DNA insertion mutants, OIST SQC for RNA-seq, ChIP-seq, and BS-seq sequencing services, Dr. Tetsuji Kakutani and Dr. Tatsuo Kanno for providing mutant seeds, Dr. Shohei Takuno for kindly sharing the list of BM genes in *A. thaliana*, OIST Infrastructure Section for technical supports in building web interface to access data, and OIST English editing service for proofreading of the manuscript.

## Author contributions

Experiments were designed by N.T.L. and H.S., and performed by Y.H., S.M., and H.S. Data analysis was performed by N.T.L., with the support of D.B. for gene expression analysis using mRNA-seq data. LTR sequences were analyzed by A.K. The manuscript was prepared by N.T.L. and H.S.

## Competing interests

The authors declare no competing interests.
