## [Peer Review File · Nature Communications]

Reviewers' comments:

Reviewer #1 (Remarks to the Author):

The study by Tu et al generates high-quality and depth CAGE data in Arabidopsis wild type and mutants compromised in silencing of heterochromatic DNA. Alternative transcriptional start sites are observed frequently in many of these mutants, with met1 often revealing the strongest effect. Using these data, EPICATS were identified that represent loci where TSSs occur in mutants that did not occur in wild type. Many of these loci harbored chromatin modifications that are indicative of silencing (H3K27me1, DNA methylation, H3K9me2) and when activated in the mutant backgrounds they contained hallmarks of active transcription (RNA POLII, chromatin accessibility, etc.). Investigation of the sequences surrounding these new TSSs show that they possess core promoter elements. The high quality nature of these data and the loss of all gene body DNA methylation in the met1 mutant allowed the investigation into whether gene body DNA methylation functions to suppress intragenic transcription, however, no strong evidence was found. Lastly, the majority of the EPICATs were located in transposons, especially those that are mobile and at times these new TSSs would lead to gene fusion events and new regulatory elements.

Overall, this is an excellent and original study that incorporates mapping of TSSs with the vast knowledge of how DNA methylation functions in plant genomes by taking advantage of mutants. The comments below are intended to strengthen this study.

Major comments:

1. The met1 rep1 is an outlier. Given this sample has three replicates I wonder how the results would change if this replicate were removed.
2. Throughout the manuscript the term "epigenetic" is used when in most cases it could be replaced with "DNA methylation" or "silencing". I think the latter more accurately describes the results. As a result, I think the title could be changed to "Suppression of spurious transcription initiation by DNA methylation in Arabidopsis".
3. The strong enrichment of new TSSs in exons in ibm1 is intriguing, but it is only briefly mentioned by the authors. Why are there so many more TSSs compared to other mutants?
4. Although the section on gene body DNA methylation is short, I believe this to be one of the most impactful results as many have hypothesized about its function with one major function being suppression of intragenic transcripts. The results show that only 28/5,826 EPICATs are located gene body methylated genes, however, the authors focused on the overlap of intraTSSs with BM genes, which is misleading. This result should be moved to Figure 4 instead of being shown in the supplementary data.
5. It is reported that there is a significant overlap between BM and intraTSSs, which is a problem. Although it is statistically significant, the enrichment is only 1.1 fold. This is a classic example of p-values being inflated due to the number of data points being used. The addition of Cohen's D would be a good way to show the strength of these signals as it will incorporate effect size. Regardless, this result is misleading as the unmethylated genes (UM) also significantly overlap intraTSSs, yet this is not presented. What the result really shows is that intraTSSs occurs in genes, but it doesn't discriminate based on the presence of gene body DNA methylation.
6. As a result of points 3 and 4, it is misleading to show anecdotal examples in Figure 4c, as they are not representative. Although they clearly show that intraTSSs can occur in BM genes, it is not a causal relationship. In the left and right panel in 4c, DNA methylation is still present in the ddm1 mutant, yet, the intraTSS is still present. Either way, similar examples could be show for UM genes and the presence of intraTSSs, as they are equally enriched in this dataset. The authors could present the distribution of the intraTSSs for the EPICATs as a gene metaplot. Are most of the new TSS originating from regions that possess DNA methylation (gene bodies) in wild-type or due they occur near the canonical TSS where DNA methylation is depleted.
7. The discussion on the gene body DNA methylation section should also be revised as the data presented are not causal, but are instead correlative.

8. Line 389, depletion of H3K36me3 has been evaluated to some extent as in reference [54], the authors explored transcriptional variation using a mutant that had no H3K36me3 and DNA methylation by using an *sdg7/8/met1* triple mutant.

Reviewer #2 (Remarks to the Author):

This manuscript explores the role of epigenetic regulation on the modulation of transcription start site (TSS) usage in Arabidopsis, by using Cap Analysis of Gene Expression (CAGE) on various mutants that compromise epigenetic control. The study provides a nice confirmation of a phenomena that is neither new nor surprising, since the effect of epigenetic control on gene expression has been known for decades, and it is obvious that it must involve the use of cryptic TSSs. The manuscript provides a few novel insights, but overall the results were already known, or could have been predicted. Important studies in Arabidopsis and other plants that directly impact the conclusions of this study were ignored. Comments that could assist the authors with a revised submission follow:

Major comments

1. It is surprising that in some instances, correlations between replicates (see *met1.1* and *met1.2*) is lower (0.77) than observed between different mutants. In fact, based on the data provided, I am suspicious about the quality of some of the replicates, including *met1.1*.
2. It is unclear how a tag cluster is defined – what is the distance allowed between CTSSs, and in which instances were CTSSs collapsed to call it a tag cluster, or what the authors call a “TSS”.
3. Given that TSSs have been previously mapped in Arabidopsis (Morton et al, 2014; Ref 36), it would have been valuable to have a correlation (ideally a figure) between the position of the TSSs identified in that study, and the TSS identified here. In my opinion, it is not enough to do this for a couple of genes, as shown in Fig S2. Moreover, the results in Fig. 1 should be compared and discussed – if pretty much the same, Fig. 1 is probably dispensable.
4. The questionable significance of the study is highlighted by the results presented between Lane 157 – 167: it is clear (as others have shown already) that CAGE-Seq has a good overlap with RNA-Seq, with the main advantage of CAGE-Seq being a more accurate prediction of TSSs (which can be implied to some extent from RNA-Seq data). The effect of epigenetic mutant on spurious transcription is well described with hundreds of RNA-Seq experiments, making it to some extent questionable what is the true contribution of this study.
5. I am confused on how the results are presented in Fig. 2. Why is the comparison done with “canonical TSS” rather than with the experimentally-determined TSS in wild type, either from this study or from the study of Morton et al? As we all know, the incorporation of TSS data into genome annotations (particularly TAIR10) is slow, yet the experimental data is available.
6. The finding that the orientation of the EPICATs is maintained is both interesting and unexpected. How is this explained? I missed what is the interpretation by the authors – it is to note that this is one of the few aspects of the study in which CAGE provides a significant advantage over RNA-Seq for the studies here described.
7. How many of the intergenic EPICATs have RNA-Seq reads that capture the transcripts from the new TSSs, and to what extent is the information on level of expression between the various mutants already available from RNA-Seq? In my opinion, this is important information to include because it would determine what is the new conceptual framework that this study is revealing, particularly when discussing the mechanisms by which EPICATs are produced.
8. The authors make no allusion to the possibility that some of the identified TSSs may correspond to recapped post-transcriptionally processed RNAs. This phenomenon is well described in humans, where up to 30% of the identified TSSs may correspond to recapping events. While this does not concern me much for the intergenic TSSs, it is something that could have a significant impact on intragenic EPICATs, particularly in *met1*.
9. It is unclear how TSSs were mapped to TEs, given that the authors focused on single-mapped reads. How was the specific TE responsible for a EPICAT identified, or was it generically assigned

to a family (given the short length of a CAGE tag, I don't see how they could be assigned to specific individuals). Yet, the really interesting biology comes when individual TEs are considered and the effect of EPICATs on adjacent genes (it is rather inconsequential to discuss the role of EPICATs in heterochromatic regions since it adds little to what is already known from RNA-Seq).

10. Lane 309-312: While it is true that the contribution of TEs to transcriptomes remains unclear, studies in maize have already shown that a large number of CAGE-identified TSSs are located within TEs. It is important that the results of this manuscript are placed on the perspective of what is known in other plants, not just Arabidopsis

11. At least twice during the manuscript the authors claim that they have produced the "first ever comprehensive maps of transcription start sites at a single base resolution". This is rather misleading as initiation of transcription and the CAGE data is far from being "single base pair" resolution – the data is clustered to accommodate CAGE reads, and the resolution is now of 10-50 bps, depending on how it is analyzed.

12. An aspect of the study that the authors failed to discuss is whether the shape of the TSS (peaked versus broad) changed in the epigenetic mutants, compared to wild-type. Arabidopsis and humans appear to have rather different TSS shapes, humans (and maize for that sake) showing more peaked transcription, while it is broad in Arabidopsis - how was this affected by the mutants?

13. How many of the EPICATs might be a consequence of POLIV or POLV transcription? Why was this not discussed?

14. The activation of new TSS sites can result in the synthesis of new proteins by the use of "cryptic" ATG codons, as initially demonstrated in maize and later in Arabidopsis – in fact, I find surprising that the CAGE study in Arabidopsis by Ushijima et al 2017 is not cited nor discussed.

Other comments

15. Lane 33: The effect of transposons on host genomes goes beyond the effects caused by mobility, as elegantly demonstrated by Barbara McClintock, and involve for example providing regulatory elements for a score of host genes.

16. Lane 39: Re-write the sentence – implies a purpose in what plants and evolution have done. I also disagree with the statements that plants are equipped with a more complex and redundant set of... it is a different set

17. Ref 33 is not appropriate in lane 92.

18. There have been a few other studies determining TSSs in plants besides Ref 34 that should be cited, including one in maize that also used CAGE

19. Lane 123: It is really irrelevant whether this study produced the largest collection of CAGE-seq data or not. I suggest to eliminate this sentence

20. It should be noted that Araport11 has been available for sometime, and provides a better annotation than TAIR10, used here

Reviewer #3 (Remarks to the Author):

The report by Tu et al studied transcriptional activation in mutants of DNA methylation. Comparative analysis of CAGE (TSS-seq), RNA-seq, bisulfite sequencing (methylome) presented solid data demonstrating gene activation in the mutants are caused by de-methylation of DNA. Experiments are well-designed, and presented information is very useful for many researchers in plant epigenetics.

However, considering that the report does not present any novel big findings, I don't recommend it for publication in Nature Communications. Rather, NAR or PLoS Genetics would be more suitable. If data revealed novel finding from pol-IV or pol-V mutants, I would welcome this report. But this is not the authors' fault, their research is fine. Very fine.

Major points

*Cryptic TSS

The authors call TSSs newly detected in the epigenetic mutants as "cryptic TSSs". Finding of this

category is the central part of the report, but after watching the presented data, I learned that they represent reactivated promoters in the mutants of DNA methylation. Their positions are 5'UTR or promoters in the gene models (Fig. 3a), close to TE (Fig. 3d and 5) and suppressed with DNA methylation (Fig. 3a, 4c, 5b), and activated in DNA methylation mutants with de-methylation (Fig. 6e and others). The whole story has been reported and is not novel.

Regarding "cryptic" promoters, there are some reports (e.g., Fobert et al, Plant J 6: 567, 1994). The reported promoters were detected by insertion of a promoter-less reporter gene (CDS). Before gene insertion, no promoter activity was detected, and after the insertion it appeared. Because position of these promoters does not locate in 5'UTR or gene upstream region, they are called as "cryptic", giving distinction from trapping of visible promoters.

I don't want to say the terminology should be exactly kept, but still I would require some non-canonical image for "cryptic". Promoters of "cryptic TSSs" identified by the authors mainly locate in 5'UTR or promoter region of gene models (Fig. 3). So they exist in expected places as promoters. Promoter elements detected from them are also normal as typical ones (Fig5. C). The presented data shows that "cryptic TSSs" come from canonical genic promoters which are suppressed by DNA methylation. I think no new name is necessary for this TSS group, because they appeared just because their promoters were activated and we don't have to assume special hidden mechanisms.

Responses to the reviewer's comments

First of all, we would like to thank the reviewers for their constructive and valuable comments on our manuscript. Below, we indicated the major changes made in the revised manuscript, and further provided point-by-point responses to the reviewers' comments.

Major changes and additional results

- As suggested by the Reviewers #1 and #2, the CAGE-seq data were re-analyzed after removing the replicate of *met1* (re-named as *met1.3*), which had lower correlations with other *met1* replicates. This removal reduced about 200 EPICATs identified in *met1* in the revised analysis (from 1663 to 1480), while the numbers of the EPICATs identified in other mutants were almost unchanged (Figure 2a). Although the main results and conclusions were not significantly affected (as shown in the revised data and manuscript), all related data were reanalyzed and reported based on the revised lists of TSSs. An EPICAT located near to the gene *AT1G23935* selected for qPCR analysis (Figure 6f), which was not present in the revised data due to its expression falling below the cut-off threshold, is presented with a notation that it was identified using all three *met1* replicates.
- As suggested by the Reviewer #1, we have replaced the Figure 4a, showing the overlap between Body-Methylated (BM) genes and genes harboring intragenic EPICATs in wild-type, by a figure showing that intragenic EPICATs were present in both BM and Non-BM genes. In addition, Supplementary Figure 7b, showing the overlap between BM genes and genes harboring intragenic EPICATs in *met1*, was moved to the main Figure 4b. A metaplot showing relative distribution of intragenic EPICATs activated in *met1* along gene body, together with the associated DNA methylation, was also added as suggested (Figure 4c). Moreover, examples of Non-BM genes harboring intragenic EPICATs in *met1* were also selected for validation, and the data were provided in Figure 4d. The whole paragraph in the section was re-written to reflect these newly added data (lines 234-251).
- In response to the Reviewers #1 and #2, two paragraphs discussing features of the intragenic EPICATs activated in *met1* (lines 253-260) and *ibm1* (lines 262-271) were added to the revised manuscript. Data supporting these discussions, including new PolII ChIP-seq data in *ibm1* background, were provided in Figure 4d, e, and Supplementary Figure 10.
- As suggested by the Reviewer #2, the TSSs identified in wild-type sample were compared to canonical TSSs annotated in Araport11 and to those identified by PEAT-seq (Morton et al., Ref. 31). The data were provided in Supplementary Figure 3.
- In response to the Reviewers #2 and #3, we further analyzed the EPICATs activated by the RdDM mutants (*pol4* and *pol5*) in more details. The new analysis lead to a

novel finding that RNAPII and PolIV exclusively function at the EPICATs regulated by the RdDM pathway. A section was added to discuss about the features of these targets (lines 274-283), and data supporting the discussion were provided in the Supplementary Figure 7c, e, 11.

- As suggested by the Reviewer #2, data showing the relationship between CAGE-seq and mRNA-seq data in characterizing expression of the EPICATs in *ddm1*, *met1*, and *suvh456* were provided in Supplementary Figure 13.
- Not related to the comments of the Reviewers, but we found a confusion in mRNA-seq data of the wild-type sample used for *de novo* transcript assembly and creating wild-type mRNA-seq tracks in the browser-track figures. Consequently, we re-analyzed and updated the data associated with mRNA-seq analysis. Accordingly, 76 gene loci exhibited differential promoter usage in *met1*. Of which, together with the gene *AT5G28842* previously selected for validation, two more were found harboring *met1*-activated EPICATs (Supplementary Data 7). Also, in order to meet the format requirements of the journal, the abstract and references were shortened compared to the previous manuscript. All the changed parts are reflected in the revised manuscript in the blue color.

Point-by-point response to the reviewers' comments

Reviewer #1: *The study by Tu et al generates high-quality and depth CAGE data in Arabidopsis wild type and mutants compromised in silencing of heterochromatic DNA. Alternative transcriptional start sites are observed frequently in many of these mutants, with met1 often revealing the strongest effect. Using these data, EPICATS were identified that represent loci where TSSs occur in mutants that did not occur in wild type. Many of these loci harbored chromatin modifications that are indicative of silencing (H3K27me1, DNA methylation, H3K9me2) and when activated in the mutant backgrounds they contained hallmarks of active transcription (RNA POLII, chromatin accessibility, etc.). Investigation of the sequences surrounding these new TSSs show that they possess core promoter elements. The high-quality nature of these data and the loss of all gene body DNA methylation in the met1 mutant allowed the investigation into whether gene body DNA methylation functions to suppress intragenic transcription, however, no strong evidence was found. Lastly, the majority of the EPICATs were located in transposons, especially those that are mobile and at times these new TSSs would lead to gene fusion events and new regulatory elements.*

Overall, this is an excellent and original study that incorporates mapping of TSSs with the vast knowledge of how DNA methylation functions in plant genomes by taking advantage of mutants. The comments below are intended to strengthen this study.

Major comments:

1. *The met1 rep1 is an outlier. Given this sample has three replicates I wonder how the results would change if this replicate were removed.*

Our response: As suggested, we have removed the replicate of *met1* (renamed as *met1.3*, lines 481-482), which had lower correlations with other *met1* replicates, and re-analyzed the data to identify the revised lists of TSSs. As shown in the Figure 2, this removal reduced the number of EPICATs in *met1* from 1663 to 1480, while the number of EPICATs in other mutants were almost unchanged. This reduction, however, did not mean that the removed EPICATs were artifacts. Rather, they were filtered out likely because their expression fell below the cut-off threshold used to identify tag clusters. One of such examples was observed at the gene *AT2G23935*, which was selected for validation by qPCR (Figure 6f). The EPICAT located near this gene was not present in the revised list of the EPICATs in *met1*. However, validated data in Figure 6f showed that this EPICAT does indeed exist, with a lower expression level than the threshold (and the previously assigned expression level). Although the main results and conclusions were not significantly changed, as shown in the revised data and manuscript, all related data were reanalyzed and reported based on the revised lists of TSSs.

2. *Throughout the manuscript the term “epigenetic” is used when in most cases it could be replaced with “DNA methylation” or “silencing”. I think the latter more accurately describes the results. As a result, I think the title could be changed to “Suppression of spurious transcription initiation by DNA methylation in Arabidopsis”.*

Our response: Of the epigenetic mutants used in our study, some are not the direct regulators of DNA methylation. For example, *suvh456* and *ibm1* directly affected the genome-wide H3K9me2. *pol4* instead affects the production of 24 nt siRNAs. In both cases, DNA methylation is affected as the consequence of epigenetic changes in the upstream targets. We, therefore, left the current title unchanged to reflect the range of epigenetic pathways that potentially regulate the activation of

EPICATs investigated in our study. Nevertheless, we have replaced the term “epigenetic mechanisms” by “repressive epigenetic modifications” wherever we found it is appropriate as suggested (for example, line 199).

3. *The strong enrichment of new TSSs in exons in ibm1 is intriguing, but it is only briefly mentioned by the authors. Why are there so many more TSSs compared to other mutants?*

Our response: Although the mechanism activating intragenic EPICATs in *ibm1* is unclear, we provided additional data supporting the discussions about their features and origin. Supplementary Figure 6d showed that *ibm1* regulated a similar number of intragenic EPICATs compared to *met1*. However, they were largely non-overlapped (Supplementary Figure 10b), suggesting that they are regulated by different mechanisms. In addition, it is unlikely that the activation of intragenic EPICATs in *ibm1* is directly regulated by the activity of IBM1, because they were not associated with increased CHG methylation in the *ibm1* background (Figure 3b, Supplementary Figure 10a). While some of these EPICATs are likely genuine TSSs (confirmed by 5'-RACE experiments shown in the newly added Figure 4e), others may correspond to 5'-capped post-transcriptionally processed mRNAs generated at the associated gene loci (Supplementary Figure 10), as the accumulation of RNAPII at these loci was not significantly affected in *ibm1* background. A paragraph summarizing these data was also added (lines 262-271).

4. *Although the section on gene body DNA methylation is short, I believe this to be one of the most impactful results as many have hypothesized about its function with one major function being suppression of intragenic transcripts. The results show that only 28/5,826 EPICATs are located gene body methylated genes, however, the authors focused on the overlap of intraTSSs with BM genes, which is misleading. This result should be moved to Figure 4 instead of being shown in the supplementary data.*

Our response: We have moved the corresponding figure to Figure 4b as suggested.

5. *It is reported that there is a significant overlap between BM and intraTSSs, which is a problem. Although it is statistically significant, the enrichment is only 1.1 fold. This is a classic example of p-values being inflated due to the number of data points being used. The addition of Cohen's D would be a good way to show the strength of these signals as it will incorporate effect size. Regardless, this result is misleading as the unmethylated genes (UM) also significantly overlap intraTSSs, yet this is not presented. What the result really shows is that intraTSSs occurs in genes, but it doesn't discriminate based on the presence of gene body DNA methylation.*

6. *As a result of points 3 and 4, it is misleading to show anecdotal examples in Figure 4c, as they are not representative. Although they clearly show that intraTSSs can occur in BM genes, it is not a causal relationship. In the left and right panel in 4c, DNA methylation is still present in the *ddm1* mutant, yet, the intraTSS is still present. Either way, similar examples could be show for UM genes and the presence of intraTSSs, as they are equally enriched in this dataset. The authors could present the distribution of the intraTSSs for the EPICATs as a gene metaplot. Are most of the new TSS originating from regions that possess DNA methylation (gene bodies) in wild-type or due they occur near the canonical TSS where DNA methylation is depleted.*

Our response to (5) and (6): We thank the Reviewer for the comments. We have replaced the previous figure by another figure showing that, in wild-type plants, intragenic TSSs (intraTSSs) were present in both BM and Non-BM genes, suggesting that intraTSSs were not specific to BM

genes (Figure 4a). Together with the intragenic TSSs activated in *met1* presented previously, we also added an example validated by 5'-RACE showing a TSS activated at a gene locus without gbM (Figure 4d). In addition, a metaplot showing relative distribution of intraTSSs along gene body, with the associated DNA methylation, was added as suggested (Figure 4c). This data showed that, intraTSSs are more present in the middle of gene units, which tend to have higher level of gbM. The corresponding paragraph was also re-written based on the revised data (lines 234-251).

7. *The discussion on the gene body DNA methylation section should also be revised as the data presented are not causal, but are instead correlative.*

Our response: The paragraph discussing about gbM was re-written as suggested (lines 404-412).

8. *Line 389, depletion of H3K36me3 has been evaluated to some extent as in reference [54], the authors explored transcriptional variation using a mutant that had no H3K36me3 and DNA methylation by using an *sdg7/8/met1* triple mutant.*

Our response: The information has been updated in the manuscript (lines 411-412) as suggested.

Reviewer #2: *This manuscript explores the role of epigenetic regulation on the modulation of transcription start site (TSS) usage in Arabidopsis, by using Cap Analysis of Gene Expression (CAGE) on various mutants that compromise epigenetic control. The study provides a nice confirmation of a phenomena that is neither new nor surprising, since the effect of epigenetic control on gene expression has been known for decades, and it is obvious that it must involve the use of cryptic TSSs. The manuscript provides a few novel insights, but overall the results were already known, or could have been predicted. Important studies in Arabidopsis and other plants that directly impact the conclusions of this study were ignored. Comments that could assist the authors with a revised submission follow:*

Major comments:

1. *It is surprising that in some instances, correlations between replicates (see *met1.1* and *met1.2*) is lower (0.77) than observed between different mutants. In fact, based on the data provided, I am suspicious about the quality of some of the replicates, including *met1.1*.*

Our response: We have removed the replicate of *met1* (renamed as *met1.3*, lines 481-482), which had lower correlations with other *met1* replicates, and re-analyzed the data for the revised lists of TSSs. As shown in the Figure 2, this removal resulted in a reduction of the EPICATs in *met1* from 1663 to 1480, while the number of EPICATs in other mutants were almost unchanged. This reduction, however, were likely due to the expression of the filtered out EPICATs fell below the cut-off threshold used to identify tag clusters. One of such examples was the EPICAT located nearby the gene *AT2G23935*, which was previously validated by qPCR (Figure 6f) although it is not listed in the revised list of the EPICATs in *met1*. Although the main results and conclusions were not significantly changed (as shown in the revised data and manuscript), all related data were reanalyzed and reported based on the revised lists of TSSs.

2. *It is unclear how a tag cluster is defined – what is the distance allowed between CTSSs, and in which instances were CTSSs collapsed to call it a tag cluster, or what the authors call a “TSS”.*

Our response: We employed the method called *paraclu* implemented in CAGEr (Ref. 78) to identify tag clusters from the CTSSs. Then tag clusters of individual samples were merged to generate a unified set of consensus clusters. Each of which was considered as a single TSS, represented by its dominant CTSS. Please refer to the Methods section for details about the method and parameters used in these steps.

3. Given that TSSs have been previously mapped in *Arabidopsis* (Morton et al, 2014; Ref 36), it would have been valuable to have a correlation (ideally a figure) between the position of the TSSs identified in that study, and the TSS identified here. In my opinion, it is not enough to do this for a couple of genes, as shown in Fig S2. Moreover, the results in Fig. 1 should be compared and discussed – if pretty much the same, Fig. 1 is probably dispensable.

Our response: As suggested, we provided additional data comparing the TSSs identified in wild-type sample of our analysis to those identified by PEAT-seq method (Ref. 31) (Supplementary Figure 3d, e). We also re-analyzed the distribution of PEAT TSSs in our framework and provided the summary in Supplementary Figure 3f. These data showed that, the two data were highly agreed, although PEAT method reported many more TSSs in exon regions (with no detailed explanation in the original report).

4. The questionable significance of the study is highlighted by the results presented between Lane 157 – 167: it is clear (as others have shown already) that CAGE-Seq has a good overlap with RNA-Seq, with the main advantage of CAGE-Seq being a more accurate prediction of TSSs (which can be implied to some extent from RNA-Seq data). The effect of epigenetic mutant on spurious transcription is well described with hundreds of RNA-Seq experiments, making it to some extent questionable what is the true contribution of this study.

Our response: While we agree that CAGE data overlap to some extent with RNA-seq data in characterizing gene expression, we provided evidences arguing against the view of the Reviewer. First, although the mutants of epigenetic pathways, such as *met1* and *ddm1*, have been reported to activate spurious transcription, genetic elements regulating such activation remain obscure since they cannot be characterized and understood in detail without the data of TSSs in the mutant backgrounds. Moreover, even the effects of epigenetic control on gene expression “must involve the use of cryptic TSSs” or “could have been predicted” as mentioned, they cannot become clear without the availability of supporting evidences. Second, it’s difficult, if not impractical, to investigate the impact of epigenetic regulation on spurious transcription at intragenic regions using RNA-seq data alone. Third, pathways exclusively functioning at the same genomic loci, as demonstrated by the data showing exclusive binding between RNAPII and PolIV at the EPICATs regulated by the RdDM pathway in the revised manuscript (Supplementary Figure 11), cannot be properly perceived without data like ours .

5. I am confused on how the results are presented in Fig. 2. Why is the comparison done with “canonical TSS” rather than with the experimentally-determined TSS in wild type, either from this study or from the study of Morton et al? As we all know, the incorporation of TSS data into genome annotations (particularly TAIR10) is slow, yet the experimental data is available.

Our response: First, there are more than one studies reporting TSSs in wild-type plants, and there were definitely technical variations among them regarding the methods used to identify the TSSs (Ref. 21, 31, and ours). Therefore, it is unreasonable to use one or the other as a benchmark data to identify the EPICATs following the approach we presented. Second, although the incorporation of TSS data into genome annotations is slow as mentioned, we realized that the number of annotated loci is still much larger than the number of canonical TSSs that could be adequately characterized by the data reported in any single study (Ref. 21 reported the largest number (~21,000) of canonical TSSs that could be characterized by their data, while ours and Ref. 31 reported a comparable number of characterizable canonical TSSs (~17,000)), leading to a higher possibility to assign regular TSSs as false positive EPICATs. These are the main reasons reference genome annotations (such as TAIR10) were used instead of data reported in an individual study. On the other hand, the same approach was already mentioned elsewhere (Ref. 26).

6. *The finding that the orientation of the EPICATs is maintained is both interesting and unexpected. How is this explained? I missed what is the interpretation by the authors – it is to note that this is one of the few aspects of the study in which CAGE provides a significant advantage over RNA-Seq for the studies here described.*

Our response: It was not clear to us what was meant by “is maintained” in this context. What can be inferred from our data is that, the transcription initiated from the EPICATs is uni-directional, not bi-directional (Supplementary Figure 6a), similar to what was described about transcription at regular TSSs in plants (Ref. 20). This is a remarkable difference in transcription at regular TSSs between plants and mammals, as shown in the previous studies (Ref. 20, 37). Such difference may be attributed to the existence of mechanisms suppressing transcription from divergent promoters (Ref. 68) or the lack of mechanisms facilitating bi-directional transcription, in plants compared to mammals. Please refer to the Discussion section for more details.

7. *How many of the intergenic EPICATs have RNA-Seq reads that capture the transcripts from the new TSSs, and to what extent is the information on level of expression between the various mutants already available from RNA-Seq? In my opinion, this is important information to include because it would determine what is the new conceptual framework that this study is revealing, particularly when discussing the mechanisms by which EPICATs are produced.*

Our response: We have provided additional data (Supplementary Data 6, 7) showing that up to 80% of the EPICATs in *met1*, *ddm1*, and *svh456* (as these mutants activated the largest numbers of EPICATs), were supported by transcripts *de novo* assembled from RNA-seq data, regarding that the EPICATs may not necessarily overlap with the supporting transcripts. In addition, expression of the EPICATs was also positively correlated with that of corresponding *de novo* assembled transcriptional units (Supplementary Figure 13).

8. *The authors make no allusion to the possibility that some of the identified TSSs may correspond to recapped post-transcriptionally processed RNAs. This phenomenon is well described in humans, where up to 30% of the identified TSSs may correspond to recapping events. While this does not concern me much for the intergenic TSSs, it is something that could have a significant impact on intragenic EPICATs, particularly in *met1*.*

Our response: We thank the Reviewer for the comments. As suggested, we have provided additional data discussing about the possibility that intragenic EPICATs in *met1* (Supplementary Figure 9c, d, e), and to some extent in *ibm1* (Supplementary Figure 10), may correspond to the 5'-end capped post-transcriptionally processed mRNAs. While we did not rule out this possibility, our data provided evidences supporting that some of these intragenic EPICATs are genuine TSSs, with validated examples by 5'-RACE experiments (Figure 4d, e). Two paragraphs summarizing these data were also added to the revised manuscript (lines 253-271).

9. *It is unclear how TSSs were mapped to TEs, given that the authors focused on single-mapped reads. How was the specific TE responsible for a EPICAT identified, or was it generically assigned to a family (given the short length of a CAGE tag, I don't see how they could be assigned to specific individuals). Yet, the really interesting biology comes when individual TEs are considered and the effect of EPICATs on adjacent genes (it is rather inconsequential to discuss the role of EPICATs on heterochromatic regions since it adds little to what is already known from RNA-Seq).*

Our response: As described in the Methods section, the TSSs were identified using only uniquely mapped reads. Because each TSS identified by CAGE data was considered to be represented by the corresponding dominant CTSS, we just simply tested whether the dominant CTSS was located in

any TE or not. Then, family information of the TEs harboring EPICATs was summarized accordingly. The only exception was the analysis of the TSSs located in the LTRs of Gypsy family, where multi-hit reads were used (Fig. 5d, Supplementary Fig. 12d). However, we did not find much difference in the results given by multi- and unique-hit reads, as many LTRs have diverged sequences even among the subfamilies.

10. Lane 309-312: While it is true that the contribution of TEs to transcriptomes remains unclear, studies in maize have already shown that a large number of CAGE-identified TSSs are located within TEs. It is important that the results of this manuscript are placed on the perspective of what is known in other plants, not just Arabidopsis.

Our response: In fact, we have noticed about the discussion of TEs which harbored CAGE-identified TSSs in the maize genome (Ref. 28). However, detail information about these TEs was not provided in the original article, as mentioned in a review paper (Ref. 56, Section 3.1.1), preventing us from further discussion on the related data. Therefore, we instead cited the review paper in our previous manuscript. As suggested, both were cited in the revised one (lines 339-340).

11. At least twice during the manuscript the authors claim that they have produced the “first ever comprehensive maps of transcription start sites at a single base resolution”. This is rather misleading as initiation of transcription and the CAGE data is far from being “single base pair” resolution – the data is clustered to accommodate CAGE reads, and the resolution is now of 10-50 bps, depending on how it is analyzed.

Our response: We have replaced the phrase “single base pair” by the phrase “high resolution” as suggested (lines 88, 239).

12. An aspect of the study that the authors failed to discuss is whether the shape of the TSS (peaked versus broad) changed in the epigenetic mutants, compared to wild-type. Arabidopsis and humans appear to have rather different TSS shapes, humans (and maize for that sake) showing more peaked transcription, while it is broad in Arabidopsis - how was this affected by the mutants?

Our response: As suggested, we have provided additional data (Supplementary Figure 6c) showing that most of the EPICATs had narrow shapes, with their widths were narrower than that of Weak Peak (WP), and narrower than or similar to that of Narrow Peak (NP) and Broad with Peak (BP) TSSs reported in Ref. 28.

13. How many of the EPICATs might be a consequence of POLIV or POLV transcription? Why was this not discussed?

Our response: We have provided additional data and discussion about the EPICATs regulated by the RdDM pathways (Supplementary Figure 7c, 11). Accordingly, a new section summarizing these data was added to the manuscript (lines 274-283).

14. The activation of new TSS sites can result in the synthesis of new proteins by the use of “cryptic” ATG codons, as initially demonstrated in maize and later in Arabidopsis – in fact, I find surprising that the CAGE study in Arabidopsis by Ushijima et al 2017 is not cited nor discussed.

Our response: As suggested, the study by Ushijima et al., 2017 was cited and discussed in the revised manuscript (Ref. 72, lines 449-452).

Other comments:

15. Lane 33: *The effect of transposons on host genomes goes beyond the effects caused by mobility, as elegantly demonstrated by Barbara McClintock, and involve for example providing regulatory elements for a score of host genes.*

Our response: The sentence was re-written which mentioned a role of TEs in providing regulatory elements to the host genomes (line 26).

16. Lane 39: *Re-write the sentence – implies a purpose in what plants and evolution have done. I also disagree with the statements that plants are equipped with a more complex and redundant set of... it is a different set.*

Our response: The sentence was re-written and the phrase “more complex and redundant” was replaced by “different” as suggested (lines 30, 33).

17. *Ref 33 is not appropriate in lane 92.*

Our response: The reference was removed as suggested.

18. *There have been a few other studies determining TSSs in plants besides Ref 34 that should be cited, including one in maize that also used CAGE.*

Our response: As suggested, the study on maize that used CAGE was cited (Ref. 28) together with Ref. 31 in the revised manuscript.

19. Lane 123: *It is really irrelevant whether this study produced the largest collection of CAGE-seq data or not. I suggest to eliminate this sentence.*

Our response: We have removed the sentence as suggested.

20. *It should be noted that Araport11 has been available for sometime, and provides a better annotation than TAIR10, used here.*

Our response: As suggested, we also compared our data with the Araport11 genome annotations, in addition to that in TAIR10. The only difference between the two results was the shift between fractions of TSSs mapped to 5' UTRs and promoter regions, which could be explained by the extension of genes' 5'-ends of Araport11 compared to those of TAIR10, while the total fraction of TSSs assigned to these two categories was unchanged (Supplementary Figure 3a, b, c). Moreover, the numbers of EPICATs identified were almost unchanged. As we observed a higher agreement between wild-type TSSs identified by CAGE data and the annotations provided by TAIR10 (Supplementary Figure 2a, b), TAIR10 annotations were used in our analysis.

Reviewer #3: *The report by Tu et al studied transcriptional activation in mutants of DNA methylation. Comparative analysis of CAGE (TSS-seq), RNA-seq, bisulfite sequencing (methylome) presented solid data demonstrating gene activation in the mutants are caused by de-methylation of DNA. Experiments are well-designed, and presented information is very useful for many researchers in plant epigenetics.*

However, considering that the report does not present any novel big findings, I don't recommend it for publication in Nature Communications. Rather, NAR or PLoS Genetics would be more suitable. If data revealed novel finding from pol-IV or pol-V mutants, I would welcome this report. But this is not the authors' fault, their research is fine. Very fine.

Major points:

- Cryptic TSS:

The authors call TSSs newly detected in the epigenetic mutants as "cryptic TSSs". Finding of this category is the central part of the report, but after watching the

presented data, I learned that they represent reactivated promoters in the mutants of DNA methylation. Their positions are 5'UTR or promoters in the gene models (Fig. 3a), close to TE (Fig. 3d and 5) and suppressed with DNA methylation (Fig. 3a, 4c, 5b), and activated in DNA methylation mutants with de-methylation (Fig. 6e and others). The whole story has been reported and is not novel.

Regarding "cryptic" promoters, there are some reports (e.g., Fobert et al, Plant J 6: 567, 1994). The reported promoters were detected by insertion of a promoter-less reporter gene (CDS). Before gene insertion, no promoter activity was detected, and after the insertion it appeared. Because position of these promoters does not locate in 5'UTR or gene upstream region, they are called as "cryptic", giving distinction from trapping of visible promoters.

I don't want to say the terminology should be exactly kept, but still I would require some non-canonical image for "cryptic". Promoters of "cryptic TSSs" identified by the authors mainly locate in 5'UTR or promoter region of gene models (Fig. 3). So they exist in expected places as promoters. Promoter elements detected from them are also normal as typical ones (Fig5. C). The presented data shows that "cryptic TSSs" come from canonical genic promoters which are suppressed by DNA methylation. I think no new name is necessary for this TSS group, because they appeared just because their promoters were activated and we don't have to assume special hidden mechanisms.

Our response:

- It was a bit surprising to us that "cryptic TSSs" reported in this study was mentioned to be "mainly located in 5' UTRs or promoter regions" as above. After revising the data, we found that this confusion was mainly due to the similarity of the colors used to mark 5' UTR and INTERGENIC TSSs. We, therefore, have changed the color marking 5' UTR TSSs to "purple", which clearly showed that cryptic TSSs identified in our study were mainly located in INTERGENIC regions. The term "cryptic TSSs" was therefore used to refer to this class of TSSs, which are regularly silenced by epigenetic mechanisms. Although we deeply understood the point raised by the Reviewer, this term has also been used in some other study (Ref. 26) with the same meaning as mentioned in this study.
- In the revised manuscript, we have further provided additional data showing that, RNAPII and PolIV exclusively function at the EPICATs regulated by the RdDM pathway (Supplementary Figure 11), which could not be elucidated without the CAGE-seq data provided in this study. We think this finding is novel and would help revise the current model, where RNAPII and PolIV were supposed to target distinct genomic territories (Ref. 50). Our data suggested that, the EPICATs regulated by the RdDM pathway likely possess distinct features compared to those of regular RdDM targets, which allow RNAPII and PolIV exclusively function at these loci. A section summarizing these data was provided in the revised manuscript (lines 274-283).

Reviewers' comments:

Reviewer #1 (Remarks to the Author):

Overall, the response is well done, with the exception of the statements (and subtitle) about the causal role of Body Methylation locally. There is no causal evidence to support this claim. The only way to do this would be to remove all of the methylation specifically at an affected locus without affecting methylation at other regions of the genome. With the exception of such an experiment and with no statistical support for enrichment, the conclusions should be modified to reflect the results.

The authors have nicely shown that intraTSSs and EPICATs are not enriched in BM compared to UM genes (4a,b). Therefore, the follow up sections/statements that BM affects some genes are inaccurate. The assumption that is being made is that loss of methylation has a DIRECT affect on these intraTSSs, whereas it could be indirect, having nothing to do with the methylation status of the gene. This is the only conclusion at this time that is supported by their current results.

This is easily addressable by modifying the subtitle and the conclusions of the last paragraphs to state that, although some BM genes show EPICATS, at this time, we don't know if this is a direct or an indirect effect of met1 mutant. Future testing using targeted demethylation could help resolve if BM is causal at these loci.

Reviewer #2 (Remarks to the Author):

The authors did a fine job addressing the numerous concerns raised by the three reviewers. The manuscript is a nice piece of work with a very impressive body of results.

My main concern with the study continues to be what I perceive as an incremental, rather than substantial, contribution to what is already known. Epigenetic mechanisms are already well known to maintain proper gene expression. The last sentence of the abstract: "Our study, therefore, sheds light on the novel role of epigenetic regulation in maintaining proper gene functions in plants by suppressing transcription from cryptic TSSs" highlights my concern - the study does not add anything new on the transcripts (which were largely described already by RNA-Seq) but shed lights on the TSSs. Whether such a contribution is sufficient for publication in Nature Comm is for the Editor to decide.

Reviewer #3 (Remarks to the Author):

Reviewer 3

I appreciate the authors for changing color in Fig. 3a. Now I have better understanding. I also understood that the topic includes selective regulation of TSS clusters for gene expression. Now I am more positive to the report.

Major points

*canonical TSSs

The authors use the word for TAIR10-annotated TSSs, and unannotated TSSs are called as "non-canonical TSSs". I say "canonical TSSs" should be ones which are supposed to transcribe whole CDS in the sense strand, regardless of if they are reported or not. Antisense, intragenic, and some other TSSs are "non-canonical". Distinction of polyA sites between canonical and non-canonical is based on gene structure as well (de Lorenzo et al, Plant Cell 29: 1262, 2017). For canonical TSSs detection, a paper in Cell (Ref 72, 2017) settles a range of 500 bp from head of a gene model

(TSS or 5' end of CDS, depending on gene models).

TSS data in TAIR10 is rather thin, and detection of new TSSs not in TAIR10 is not very surprising.

*Figure 2a

Distinction between reported and unreported TSSs is not very important. I suggest they are mixed and EPICATs are defined according to only comparison between wt and mt data.

If unreported TSSs are still to be detected, more comprehensive TSS data (Ref 21, 2017) than TAIR10 should be used.

*Pol IV and V

Mixing of EPICATs detected from methylation mutants and from pol IV/V mutants is not recommended. They can be divided into two subgroups, or two reports. Separation would give sharper results in motif search and some other analyses.

The above three points affect definition of EPICATs and would change results considerably.

Minor points

*Cryptic TSSs

If selective regulation among multiple TSS clusters connected to a gene model is observed, this terminology is acceptable.

Responses to the reviewer's comments

We would like to thank the reviewers for their valuable comments on our revised manuscript. Below, we indicated the changes made in the (second) revised manuscript and provided point-by-point responses to the reviewers' comments. All the changed parts are reflected in the blue color in the (second) revised manuscript.

Point-by-point response to the reviewers' comments

Reviewer #1: *Overall, the response is well done, with the exception of the statements (and subtitle) about the causal role of Body Methylation locally. There is no causal evidence to support this claim. The only way to do this would be to remove all of the methylation specifically at an affected locus without affecting methylation at other regions of the genome. With the exception of such an experiment and with no statistical support for enrichment, the conclusions should be modified to reflect the results.*

The authors have nicely shown that intraTSSs and EPICATs are not enriched in BM compared to UM genes (4a,b). Therefore, the follow up sections/statements that BM affects some genes are inaccurate. The assumption that is being made is that loss of methylation has a DIRECT affect on these intraTSSs, whereas it could be indirect, having nothing to do with the methylation status of the gene. This is the only conclusion at this time that is supported by their current results.

This is easily addressable by modifying the subtitle and the conclusions of the last paragraphs to state that, although some BM genes show EPICATS, at this time, we don't know if this is a direct or an indirect effect of met1 mutant. Future testing using targeted demethylation could help resolve if BM is causal at these loci.

Our response:

We thank the Reviewer for the detailed comments and suggestions. Accordingly, we have changed the subtitle of the corresponding section (Lines 232-233), and the conclusions of the first paragraph (Lines 248-250) as suggested. The third paragraph of the Discussion section was also updated by removing inappropriate sentences (Lines 402-413).

Reviewer #2: *The authors did a fine job addressing the numerous concerns raised by the three reviewers. The manuscript is a nice piece of work with a very impressive body of results.*

My main concern with the study continues to be what I perceive as an incremental, rather than substantial, contribution to what is already known. Epigenetic mechanisms are already well known to maintain proper gene expression. The last sentence of the abstract: "Our study, therefore, sheds light on the novel role of epigenetic regulation in maintaining proper gene functions in plants by suppressing transcription from cryptic TSSs" highlights my concern - the study does not add anything new on the transcripts (which were largely described already by RNA-Seq) but shed lights on the TSSs. Whether such a contribution is sufficient for publication in Nature Comm is for the Editor to decide.

Our response:

We thank the Reviewer for the positive judgement on our revised manuscript. We think the TSS information of epigenetic mutants in our reports would further advance the understanding of epigenetic regulation of transcription of genes and TEs, which would also complement the knowledge relying only on RNA-seq data.

Reviewer #3: *I appreciate the authors for changing color in Fig. 3a. Now I have better understanding. I also understood that the topic includes selective regulation of TSS clusters for gene expression. Now I am more positive to the report.*

Major points:

- Canonical TSSs:

The authors use the word for TAIR10-annotated TSSs, and unannotated TSSs are called as "non-canonical TSSs". I say "canonical TSSs" should be ones which are supposed to transcribe whole CDS in the sense strand, regardless of if they are reported or not. Antisense, intragenic, and some other TSSs are "non-canonical". Distinction of polyA sites between canonical and non-canonical is based on gene structure as well (de Lorenzo et al, Plant Cell 29: 1262, 2017). For canonical TSSs detection, a paper in Cell (Ref 72, 2017) settles a range of 500 bp from head of a gene model (TSS or 5' end of CDS, depending on gene models).

TSS data in TAIR10 is rather thin, and detection of new TSSs not in TAIR10 is not very surprising.

Our response:

In our analysis, CAGE-detected TSSs within 180 bp of the TSSs annotated by TAIR10 were categorized as “annotated”, which are in accordance with the criteria mentioned by the reviewer. This was derived from the observation that, more than 90% of the CAGE-detected TSSs mapped to promoters (1 kb upstream) and 5' UTRs of the canonical TSSs were within this distance (False Discovery Rate = 0.1, Supplementary Figure 2b). Furthermore, as previously suggested by the second Reviewer, in addition to genome annotations in TAIR10, the latest version of genome annotations of *Arabidopsis thaliana*, i.e. Araport11, was also used to compare with the TSSs detected by our CAGE-seq data and to identify EPICATs (Supplementary Figure 3a, b, c). Both resulted in a similar number of EPICATs (Supplementary Figure 5), demonstrating the robustness of our analysis.

- Figure 2a:

Distinction between reported and unreported TSSs is not very important. I suggest they are mixed and EPICATs are defined according to only comparison between wt and mt data.

If unreported TSSs are still to be detected, more comprehensive TSS data (Ref 21, 2017) than TAIR10 should be used.

Our response:

The EPICATs in the Figure 2a were indeed identified by comparing between wild-type and mutants, and then classified based on their association with the genome annotations as described in the above response. Ignoring the difference between annotated and non-annotated TSSs can lead to the misclassification of regular TSSs, which are silenced in wild-type and activated in the mutants as exemplified at *FWA* and *SDC* loci (Figure 1c), as EPICATs. We have also clearly explained in the previous responses to the second Reviewer (in the first revised manuscript) the reasons why we did not use the TSSs reported in any individual study as a benchmark data for identifying EPICATs. The first one is technical variations regarding the methods used to identify TSSs, and the second one is a higher possibility of assigning regular TSSs as false positive EPICATs. Please refer to our previous Response Letter (Reviewer #2, question 5) for the details.

- *Pol IV and V:*

Mixing of EPICATs detected from methylation mutants and from pol IV/V mutants is not recommended. They can be divided into two subgroups, or two reports. Separation would give sharper results in motif search and some other analyses.

The above three points affect definition of EPICATs and would change results considerably.

Our response:

The analyses of EPICATs in the mutants (Supplementary Figure 6, 7, 8), including the motif search (Supplementary Figure 8), were performed only on the EPICATs identified in each mutant, not on the “mixed EPICATs”. Moreover, the EPICATs activated in *pol4* and *pol5* were shown to have narrow shapes (Supplementary Figure 6c), and the motif search was conducted on 100 bp regions centering around their dominant CTSSs regardless of the shapes of the corresponding consensus tag clusters (please see Methods section for detail). These facts eliminate the speculation that “*Separation would give sharper results in motif search and some other analyses*”. We also found this comment inconsistent with the previous comment by the Reviewer, as we performed additional analyses of *pol4/5* mutants in the revised manuscript to respond to the previous comment: “*If data revealed novel finding from pol-IV or pol-V mutants, I would welcome this report.*”

Minor points

- *Cryptic TSSs:*

If selective regulation among multiple TSS clusters connected to a gene model is observed, this terminology is acceptable.

Our response:

We have demonstrated that the activation of the EPICATs indeed leads to the formation of fusion transcripts at several gene loci (Figure 6b, c, Supplementary Figure 14a, b), consistent with the usage of the term “Cryptic TSSs”.